# Nutritional Variation on Sequentially Harvested Shoots and Fruits of *Lagenaria siceraria* Landraces

**DOI:** 10.3390/plants13111475

**Published:** 2024-05-27

**Authors:** Lungelo Given Buthelezi, Sydney Mavengahama, Julia Sibiya, Charmaine Nontokozo Mchunu, Nontuthuko Rosemary Ntuli

**Affiliations:** 1Department of Agriculture, Faculty of Science, Agriculture and Engineering, University of Zululand, KwaDlangezwa 3886, South Africa; 2Food Security and Safety Area, Faculty of Natural and Agricultural Science, North-West University, Mmabatho 2745, South Africa; sydney.mavengahama@nwu.ac.za; 3School of Agriculture, Earth and Environmental Sciences, University of KwaZulu-Natal, Pietermaritzburg 3201, South Africa; sibiyaj@ukzn.ac.za; 4KwaZulu-Natal Department of Agriculture & Rural Development, Soil Fertility and Analytical Services, 01 Cedara Road, Pietermaritzburg 3200, South Africa; charmaine.mchunu@kzndard.gov.za; 5Department of Botany, Faculty of Science, Agriculture and Engineering, University of Zululand, KwaDlangezwa 3886, South Africa; ntulir@unizulu.ac.za

**Keywords:** nutrients, variation, landraces, *Lagenaria siceraria*

## Abstract

*Lagenaria siceraria* (Molina) Standley, a member of the Cucurbitaceae family, is valued for its medicinal and nutritive properties. The nutrient status of sequentially harvested shoots and fruits at various growth stages in *L. siceraria* has not been documented to date. This study aimed to compare the nutritional status of *L. siceraria* landrace shoots and fruits harvested at different maturity stages. Micronutrients, macronutrients, and proximate composition of shoots and fruits were determined using inductively coupled plasma–optical emission spectrometry (ICP-OES). Data were subjected to multivariate statistical analysis. The nutrient attributes differed significantly (*p* < 0.05) within and among landraces at different growth stages. Correlation of nutritional traits was primarily based on shared absorption sites and comparable chemical composition. The first five principal components in shoots and fruits had 90.218 and 89.918% total variability, respectively. The micronutrients Ca, Mg, K, P, and N in shoots and the macronutrients Fe, Zn, Cu, and Al in fruits were the main contributors to variability. The biplot and dendrogram clustered landraces with comparable nutrient values. Shoot traits classified landraces into three major clusters, where clusters I and II grouped landraces with superior and inferior Mg, P, K, N, Fe, K/Ca+Mg, ADL, and protein levels at 42–63 DAS. Cluster III consisted of landraces with more Ca, Na, Mn, Zn, and Cu. However, the fruit nutrient status classified landraces into two major clusters. Cluster I comprised landrace KSC (as a singlet) with the highest Ca, P, N, Mn, Fe, Zn, and Cu contents at various stages of growth (7–21 DAA). Cluster II had landraces with higher K, C/N ratio, Na, moisture content, ash, protein, and fat. The nutritional status of shoots and fruits determined at different stages of growth is essential for selecting the best harvest time and landrace(s) for required daily nutrient intake.

## 1. Introduction

*Lagenaria siceraria* (Molina) Standley is a hardy crop that thrives in warmer regions of the planet for its various applications [1]. With the lack of animal nutrient accessibility, underutilized crops such as *L. siceraria*, whose shoots, leaves, flowers, juvenile and mature fruits, and seeds can be consumed as a vegetable relish [2], are gaining research popularity due to their potential as a prominent dietary supplement for human consumption [3].

Minerals are inorganic elements, some of which are vital nutrients [4]. The minerals obtained from the consumption of plant parts are regarded as essential nutrients for human health [1]. They are clustered into two distinct groups: macronutrients (K, P, Mg, Na, and Ca) and micronutrients (Mn, Zn, Fe, and Cu), which are important constituents in a myriad of biological pathways for plant and human growth [5]. For humans, the lack of these minerals can result in organ failure, metabolic disorders and, subsequently, death [1]. Thus, they must be obtained from food; natural plant foods such as fruits and vegetables are the primary sources of these mineral elements [6].

*Lagenaria siceraria* is abundant in some of these essential nutrients for mankind’s dietary requirements for survival [3]. Its leaves contain essential nutrients and agronomic indicators such as moisture content, carbohydrates, protein, mineral elements, and fiber [7]. The leaves also contain sufficient amounts of phosphorus (P), calcium (Ca), magnesium (Mg), zinc (Zn), copper (Cu), iron (Fe), and manganese (Mn) [8]. Similar to the leaves, *L. siceraria* fruits also contain essential nutrients such as moisture content, ash, protein, lipids, and carbohydrates [9]. It has neutral detergent fiber (NDF), acid detergent fiber (ADF), and vitamin C [10]. *Lagenaria siceraria* fruits also contain essential mineral elements such as iron, copper, silver (Ag), manganese, zinc, cobalt (Co), and lead (Pb) [4,11]. They also contain phosphorus, potassium, magnesium, calcium, and sodium [12].

The nutritional composition of *L. siceraria* is among the most researched topics in recent times around the world due to the availability and quantities of its nutritional composition as well as its various uses, supporting it as an alternative nutritious crop. However, currently, no study has investigated the variation in nutrient composition in the shoots and fruits of *L. siceraria* across different growth stages. Hence, a study on the variation in nutrient content in shoots and fruits across different growth stages will be beneficial for harvesting the plant parts at their most nutritionally valuable growth stage, thus allowing farmers, scientists, and fellow researchers to schedule the crop accordingly for market readiness. In the current study, we documented the nutrient content of shoots and fruits of *L. siceraria* landraces at different growth stages, making it the most in-depth and distinctive study to date worldwide.

## 2. Results

The description of fruit and seed traits of the nineteen landraces of *Lagenaria siceraria* from different agro-climatic regions in northern KwaZulu-Natal and Limpopo, South Africa, can be found in Table 1. Fruit and seed morphological attribute variations can be seen in Figure 1 and Figure 2, respectively.

### 2.1. Macronutrient Composition

The macronutrient composition of landraces differed significantly (*p* < 0.05) in shoots and fruits harvested at the same stage of development and among each landrace at different stages of development (Table 2). At 49 DAS, KSP had higher calcium than at 42 and 56 DAS. Furthermore, landrace KRI had higher calcium levels at 56 DAS than at 42, 49, and 63 DAS. Again, the calcium content of BG-19, BG-26, BG-27, BG-100/GC, and ESC was greater at 63 DAS than at 42 DAS.

Among the fruits assessed, landraces ESC, KSC, and NSRP had the highest calcium concentration at 7, 14, and 21 days after anthesis (DAA), respectively. Variation in calcium content within landraces across different growth stages recorded that BG-100/GC and NRC produced higher calcium levels at 21 DAA compared with 7 and 14 DAA. Landraces KSC and NSRP also produced more calcium at 14 and 21 DAA than at 7 DAA. Further, the following landraces had more calcium content at 7 DAA compared with 14 and 21 DAA: DSI, KSP, and NSRC, and at 21 DAA (ESC). Landrace RSP showed a different trend, with more calcium at 7 and 14 DAA than at 21 DAA. 

Landrace KRI contained the most magnesium at all stages of growth compared with other landraces (Table 2). Landrace NqSC produced more magnesium content at 42 DAS than at 56 DAS. Landrace KRI had more magnesium at 56 DAS than at 49 DAS, whereas BG-27 recorded higher magnesium content at 63 DAS than at 42, 49, and 56 DAS. Magnesium content in fruits remained relatively similar among all landraces at 7 and 14 DAA. However, at 21 DAA, landrace BG-27 produced the most magnesium of all landraces across all growth stages (Table 2). 

The highest potassium content was recorded in landrace KRI at 42 DAS (Table 2). However, all landraces had similar potassium content at other stages of growth (49, 56, and 63 DAS). Potassium content also remained similar within each landrace as the shoots grew. In fruits, landraces DSI and RSP exhibited the highest potassium content among all landraces at 7 and 14 DAA (Table 2). However, at 21 DAA, potassium levels were relatively similar among all landraces. Variation in potassium content across growth stages within landraces demonstrated that DSI and NqRC exhibited higher potassium levels at 7 DAA compared with 14 and 21 DAA. Additionally, RSP showed higher potassium content at 14 DAA than at 21 DAA.

When compared with other landraces, KRI recorded the most phosphorus content at 42, 56, and 63 DAS, whereas BG-26 and BG-19 had the highest phosphorus at 49 and 63 DAS, respectively (Table 2). Comparison of phosphorus content at different harvest days within each landrace showed that landraces KSC and KSP had higher phosphorus levels at 42 and 49 DAS than at 63 DAS, whereas BG-26 also had more phosphorus at 42 and 49 DAS than at 56 and 63 DAS. At 7 and 14 DAA, the fruits of KRI and KSC produced the highest phosphorus content among the landraces (Table 2). Additionally, at 21 DAA, landraces BG-24, BG-27, DSI, and NqSC exhibited higher phosphorus levels compared with KRI and KSP. The phosphorus content of the landraces varied across growth stages. Landrace BG-24 had higher phosphorus levels at 7 and 21 DAA than at 14 DAA, while DSI and NqRC contained more phosphorus at 14 and 21 DAA than at 7 DAA. Landraces ESC and NqSC produced more phosphorus at 7 and 14 DAA than at 21 DAA, and at 21 down to 7 DAA, respectively. More phosphorus was recorded in landraces KSC and KRI at 14 DAA than at 7 DAA, while KSP produced more phosphorus at 7 DAA than at 14 and 21 DAA. Landraces NSRC and NSRP produced higher phosphorus content at 14 DAA than at 7 and 21 DAA.

The fruit nitrogen content did not vary among landraces at 7 and 21 DAA (Table 2). However, at 14 DAA, KSC contained the highest nitrogen content of them all. The nitrogen content of KRI was higher at 7 DAA than at 14 and 21 DAA. Again, KSC had more nitrogen at 14 DAA than at 7 and 21 DAA. Furthermore, the carbon content in fruits remained similar among landraces and across different harvest periods (Table 2). At 7 DAA, landrace BG-31 had a higher carbon-to-nitrogen ratio (C/N ratio) than BG-24, BG-100/GC, KRI, KSP, and NSRP (Table 2). At 14 and 21 DAA, landrace KRI demonstrated the highest C/N ratio. The C/N ratio varied within landraces across different stages of growth. Landraces BG-24, ESC, and NSRP had a higher C/N ratio at 14 and 21 DAA than at 7 DAA (Table 2). Furthermore, landraces BG-100/GC, KSC, KSP, NRC, NqRC, and RSP showed a higher C/N ratio at 21 DAA than at 7 and 14 (KSC and NqRC) DAA. Additionally, BG-100/GC, DSI, KSP, NRC, NSRC, and RSP had a higher C/N ratio at 7 and 21 DAA than at 14 DAA. Landrace KRI had the highest C/N ratio at 14 DAA, followed by 21 DAA, and the lowest at 7 DAA.

### 2.2. Micronutrient Composition

The micronutrient content of landraces varied significantly (*p* < 0.05) within each growth phase and as the shoots were maturing and vining from 42 to 63 DAS (Table 3). Landraces BG-19 and BG-100/GC at 42 DAS; KRI and NqSC at 49 DAS; BG-19, BG-26, DSI, ESC, KSP, and NqSC at 56 DAS; and BG-19, ESC, and KSP at 63 DAS produced more sodium content than other landraces. The majority (85%) of the landraces had higher sodium levels at 56 DAS than at 42 and 49 DAS, with very few exceptions. In fruits, landraces DSI, BG-31, and KSC had the highest sodium content at 7, 14, and 21 DAA, respectively (Table 3). Further, the landraces displayed variations in sodium concentrations across different harvest periods. Landrace BG-100/GC had higher sodium levels at 14 DAA than at 7 and 21 DAA, while DSI had higher sodium levels at 7 DAA than at 14 and 21 DAA. Landraces BG-24, ESC, KRI, NqRC, and NRC also produced more sodium at 21 DAA than at 7 and 14 DAA. Furthermore, landraces BG-31, KSC, NSRP, and RSP had more sodium at 14 and 21 DAA than at 7 DAA. Additionally, KSP produced more sodium at 7 and 21 DAA than at 14 DAA.

Landrace BG-19 had higher manganese content than other landraces at all stages of growth, with very few exceptions (Table 3). Again, the manganese content did not vary in 77% of the landraces across varying stages of growth. However, landrace BG-26 showed higher levels of manganese at 56 and 63 DAS than at 42 and 49 DAS. Furthermore, BG-100/GC and KSC had more manganese content at 63 DAS than at 42 DAS as well as 42 and 49 DAS, respectively. Landraces ESC, KSC, and BG-31 had the highest fruit manganese content at 7, 14, and 21 DAA, respectively (Table 3). The variation in manganese concentrations within the same landrace across different harvest periods demonstrated that landraces BG-24, BG-100/GC, ESC, and KRI had more manganese at 7 DAA than at 14 and 21 DAA. Landraces KSC and NqRC produced more manganese content at 14 DAA than at 7 and 21 DAA, while in landraces KSP and NRC manganese was higher at 7 and 21 DAA than at 14 DAA.

Landrace BG-100/GC had the highest iron content at 42, 49, and 56 DAS, whereas landrace KSC had the highest iron levels at 63 DAS compared with other landraces (Table 3). The shoot iron content of BG-19, KSP, NqSC, and KRI was higher at 42 DAS than at 49 DAS and then increased at 56 and 63 DAS. Furthermore, landraces BG-26, BG-27, BG-81, and BG-100/GC had more iron at 42 DAS than at 49 DAS, then increased at 56 DAS before dropping at 63 DAS (Table 3). Again, DSI and KSC exhibited steadily rising iron levels from 42, 49, and 56 to 63 DAS. The highest fruit iron content at 7, 14, and 21 DAA was recorded in landraces ESC, KSC, and NRC, respectively (Table 3). The iron content also varied across fruit growth stages. Landraces BG-24 and KSC had higher iron levels at 14 DAA than at 7 DAA, whereas landraces BG-31, BG-100/GC, and ESC produced more iron as fruits matured from 7 to 21 DAA. Landraces DSI, NqRC, NSRC, and NSRP produced more iron at 14 DAA than at 21 DAA, while landraces NqSC and NRC had more iron at 21 DAA than at 7 and 14 DAA. Landrace BG-27 contained more iron at 7 DAA than at 14 and 21 DAA, while landrace RSP had more iron at 21 DAA than at 7 DAA (Table 3).

The zinc content in shoots of landrace BG-19 at 42 DAS; landraces BG-19, BG-24, BG-26, and BG-80 at 49 DAS; landraces BG-19 and BG-26 at 56 DAS; and landrace BG-81 at 63 DAS was higher than that of other landraces (Table 3). Many landraces (69%) had higher zinc content at 63 DAS than at all other development phases, with few exceptions. Furthermore, BG-26, BG-27, and KSP had more zinc at 63 DAS than at 42 (BG-26) and 49 DAS. Moreover, BG-100/GC had higher zinc levels at 56–63 DAS than at 42–49 DAS, but KSC had more zinc at 49, 56, and 63 DAS than at 42 DAS. 

Landraces ESC and DSI at 7 DAA, KSC at 14 DAA, and BG-24 and BG-27 at 21 DAA produced fruits with the highest zinc content (Table 3). Variability in the zinc content of landraces across different stages of growth also demonstrated that landrace BG-24 produced fruits with higher zinc levels at 7 and 21 DAA than at 14 DAA. Landraces BG-100/GC and NSRC also had more zinc at 7 and 14 DAA than at 21 DAA. Furthermore, fruits of DSI and ESC produced the highest zinc content at 7 DAA, which progressively decreased as the fruits matured from 14 to 21 DAA. Again, KRI had more zinc at 7 DAA than at 14 and 21 DAA, whereas NqSC had more zinc at 14 and 21 DAA than at 7 DAA. Moreover, KSC fruits produced more zinc at 14 DAA than at 7 and 21 DAA. Landrace NqRC had the highest zinc content at 14 DAA, followed by 21 DAA, and the least at 7 DAA.

Landraces BG-100/GC and BG-19 had higher copper content than other landraces at 42 and 63 DAS, respectively (Table 3). However, copper content did not differ between landraces at 49 and 56 DAS. Again, 77% of landraces did not show variation in copper levels across all development stages. Landraces BG-24 and BG-27 exhibited higher copper content at 63 DAS than at 42 DAS, whereas BG-19 and BG-27 contained more copper at 63 DAS than at 49 DAS.

The fruits of landraces ESC, KSC, and NSRP produced the highest fruit copper content at 7, 14, and 21 DAA, respectively (Table 3). Landraces also showed variation in copper levels across varying stages of growth. Landrace BG-24 had more copper at 7 and 21 DAA than at 14 DAA. Again, landraces BG-100/GC, ESC, KRI, and NRC had more copper at 7 DAA, which progressively decreased with fruit maturity from 14 to 21 DAA. Furthermore, DSI, NSRP, and RSP produced higher copper levels at 21 DAA than at 7 and 14 DAA. Again, NqRC had the highest copper content at 21 DAA, followed by 14 DAA, and the lowest at 7 DAA. Additionally, landraces NqSC and NSRC had more copper at 14 and 21 DAA than at 7 DAA. Fruit aluminum content did not vary among all landraces and across all growth stages (Table 3).

### 2.3. Proximate Composition

Proximate composition varied significantly (*p* < 0.05) among landraces within each stage of growth and as the shoots and fruits matured from 42 to 63 DAS and from 7 to 21 DAA, respectively (Table 4). Many landraces had similar moisture content compared with others in each stage of growth, as was clearly presented at 56 DAS. However, landraces KSC and NqSC at 42 DAS, DSI at 49 DAS, and NqSC at 63 DAS had sappier shoots than other landraces, including those of BG-27, at 42 DAS. Apart from landrace BG-26, which showed higher moisture content at 42–49 DAS than at 56–63 DAS, almost all landraces (97%) had similar moisture content at all growth stages. The majority (63%) of landraces produced sappier fruits at 7–14 DAA than at 21 DAA. The highest moisture content was recorded in KSP, NSRC, and BG-100/GC at 7, 14, and 21 DAA, respectively. Apart from BG-100/GC, which had the sappiest fruits at 21 DAA, followed by 14 DAA, and the least at 7 DAA, the majority (67%) of landraces had more succulent fruits at 7 DAA. However, landraces ESC, KRI, NqRC, NqSC, and NSRC had fruits with higher moisture content at 7 and 14 DAA than at 21 DAA. Landrace KRI at 42 and 56 DAS, as well as KSP at 63 DAS, had greater acid detergent fiber (ADF) content than other landraces (Table 4). However, landraces did not differ in their ADF content at 49 DAS. Additionally, the majority of landraces (85%) had more ADF content at 63 DAS than at the earlier growth stage. In fruits, the highest acid detergent fiber (ADF) concentration was recorded in DSI, NSRP, and ESC fruits at 7, 14, and 21 DAA, respectively (Table 4). The majority (78%) of landraces that showed variations in ADF content across different growth stages had more ADF at 21 DAA than at 7 and 14 DAA (Table 4). Landraces ESC, KSC, NSRC, and NSRP had the highest ADF at 21 DAA, followed by 14 DAA, and the lowest at 7 DAA. Further, landraces KRI and NqSC had higher ADF at 21 DAA than at 14 DAA. Again, landrace RSP produced more ADF at 21 DAA than at 7 DAA. Landraces BG-100/GC and KRI at 42 DAS; KRI at 49 DAS; BG-100/GC, DSI, and KRI at 56 DAS; and KSP at 63 DAS exhibited higher shoot neutral detergent fiber (NDF) content than other landraces (Table 4). Landraces DSI, NSRP, and ESC had the highest fruit neutral detergent fiber (NDF) content than other landraces at 7, 14, and 21 DAA, respectively (Table 4). 

At 42, 49, and 56 DAS, the highest acid detergent lignin (ADL) content was recorded in landrace KRI (Table 4). Moreover, landraces BG-19, KRI, and NqSC had higher ADL content than all other landraces at 63 DAS. The ADL content of landraces BG-19, BG-24, BG-26, BG-80, BG-81, BG-100/GC, ESC, KSP, and NqSC was higher at 63 DAS than at 42–56 DAS. Furthermore, at 56 DAS, KRI produced more ADL than at 42, 49, and 63 DAS, where the content recorded at 42 and 63 DAS was higher than at 49 DAS. Again, at 42, 56, and 63 DAS, KSC had more ADL than at 49 DAS. Moreover, at 63 DAS, BG-27 had more ADL than at 42 DAS. Landraces BG-27, ESC, KSC, and RSP produced more ash at 14 DAA than at 7 DAA. Additionally, landraces BG-100/GC and NSRP had more ash at 21 DAA than at 7 DAA. 

The protein content was consistent across all landraces and at different stages of shoot growth (Table 4). Protein content was relatively similar among all landraces and at different stages of fruit growth, with very few exceptions (Table 4). Landraces NSRP and BG-100/GC produced fruits with the highest protein content at 7 and 14 DAA. However, the protein content was similar among all landraces at 21 DAA. The majority (63%) of landraces showing variations in protein content recorded their highest protein content at 14 DAA. Landraces KRI, KSC, and NSRP had more protein at 7 DAA than at 21 DAA. Further, landraces BG-100/GC, ESC, KSP, NqRC, and NRC produced more protein at 14 than 21 DAA.

Landrace BG-27 at 42 DAS, ESC at 49 and 56 DAS, and BG-27 and ESC at 63 DAS contained the most carbohydrates of any landrace (Table 4). The carbohydrates in landraces BG-24, BG-27, BG-100/GC, and DSI were higher at 42 DAS than at 49–63 DAS. Furthermore, at 42–49 DAS, BG-26, BG-80, and KSC had more carbohydrates than at 56–63 DAS. Again, carbohydrate levels were higher at 49 DAS in BG-19, BG-81, ESC, and KRI than at 42, 56, and 63 DAS. Moreover, KSP had more carbohydrates at 42–56 DAS than at 63 DAS, whereas NqSC produced more carbohydrates at 56 DAS than at 42, 49, and 63 DAS. Landrace RSP, NSRP, and NRC produced fruits with the highest fat content at 7, 14, and 21 DAA, respectively (Table 4). A total of 60% of landraces produced more fat at 14 and 21 DAA than at 7 DAA. Landraces BG-24, BG-100/GC, ESC, and RSP had higher fat content at 7 DAA than at 14 and 21 DAA. Higher fat content was recorded at 14 DAA than at 7 DAA for KSP and NSRP and 21 DAA for KRI and KSP. Moreover, BG-31, NqSC, NRC, and NSRC had more fat at 21 DAA than at 7 DAA. However, landraces DSI produced more fat at 21 than 14 DAA.

### 2.4. Correlation among Nutritional Components of Lagenaria siceraria Shoots and Fruits 

In the *L. siceraria* shoots, calcium correlated positively with Mg and ADL but negatively with the K/Ca+Mg ratio (Table 5). A positive correlation was recorded among Mg, K, P, N, ADL, and proteins. However, Mg and K correlated negatively with Fe. Nitrogen and proteins further correlated positively with the moisture content. Sodium was positively correlated with manganese and carbohydrates. Manganese further associated positively with zinc. Iron was negatively associated with ADL. ADF had a positive correlation with NDF, and they both correlated negatively with carbohydrates. ADL correlated positively with proteins but negatively with carbohydrates. In the fruits, a positive correlation was recorded among calcium, ADF, and NDF (Table 6). Potassium had a positive correlation with ash. Carbon, copper, and potassium correlated positively with each other, while carbon further correlated positively with N, Zn, and protein. Nitrogen correlated positively with Fe, Zn, Al, and protein but negatively with the C/N ratio. C/N ratio further correlated negatively with protein, while sodium was associated negatively with moisture content. A positive correlation was also evident among Al, Mn, Fe, Zn, and protein.

### 2.5. Principal Component Analysis 

The first five informative principal components (PC1–5) accounted for 90.218% of the total variability (Table 7). The first principal component (PC1), which contributed 42.312% to the total variation, was positively associated with Ca, Mg, K, P, N, ADL, and proteins but negatively with Fe and carbohydrates.

Further, PC2, with a variability of 17.594%, was positively correlated with ADF and NDF but negatively with Mn. Again, PC3, which accounted for 12.050% of the total variation, was positively associated with Zn and Cu. Furthermore, PC4, which contributed 10.058% to the total variability, correlated positively with Na and moisture content, while PC5, which accounted for 8.204% of the total variability, correlated positively with the K/Ca+Mg ratio. Similarly for fruits, the first five principal components (PC1–5) amounted to 89.918% total variability (Table 8). The first principal component (PC1), with a total variation of 37.069%, was positively correlated with N, Fe, Zn, Cu, Al, ADF, and protein but negatively with the C/N ratio. Furthermore, PC2 with a variability of 18.781% was positively associated with Ca, K, and ash but negatively with P. Moreover, PC3 responsible for 18.042% variability correlated positively with Mn and moisture content but negatively with Mg, C, and Na.

### 2.6. Cluster Analysis

In a biplot for shoots, Ca, Mg, K, P, N, ADL, and protein correlated positively only with the first principal component (PC1) (Figure 3a). Furthermore, Fe, K/Ca+Mg, and Cu were positively correlated only with PC2, whereas NDF, ADF, and moisture content had a positive correlation with both PC1 and PC2. Moreover, carbohydrates, Zn, Mn, and Na negatively correlated with both PC1 and PC2. Landraces were grouped into three clusters in a biplot, where BG-100/GC and KRI each formed singletons in clusters III and II, respectively. All other landraces were grouped into cluster I. In fruits, Ash, K, Ca, ADF, NDF, Mn, Na, fat, and Al correlated positively with both PC1 and PC2 (Figure 3b). The N/C ratio was positively associated with PC2 and negatively with PC1. Furthermore, Fe, Zn, protein, N, moisture content, C, Cu, Mg, and P correlated positively with PC1 but negatively with PC2. Landraces were grouped into four clusters in a biplot, where KSC, BG-27, and RSP each formed singletons in clusters II, III, and IV, respectively. All other landraces formed cluster I. A hierarchical cluster analysis of shoot nutrients classified landraces into three major clusters (clusters I–III) (Figure 4a). Clusters I and II were made up of landraces KRI and BG-100/GC, respectively. Cluster III was subdivided into two sub-clusters that had landraces BG-24, NqSC, BG-27, and ESC assigned to sub-cluster IIIa and landraces KSP, BG-19, BG-80, DSI, BG-26, and BG-81 to sub-cluster IIIb. Furthermore, the hierarchical cluster analysis of fruit nutrient traits (Figure 4b) grouped landraces into two major clusters (clusters I and II). Cluster I was made up of KSC, while cluster II was further subdivided into subclusters, with cluster IIa consisting of ESC and cluster IIb made up of all the remaining landraces.

## 3. Discussion

In our study, the differences in mineral elements and proximate composition changed a lot among landraces and as they grew and matured. These differences were most likely imposed by the degree of plant age, fruit maturation, different areas of origin, and genotype diversity of the landraces studied [14]. The findings also support the sink power of mineral and proximal content in fruits, which accumulate during various stages of maturation [15].

### 3.1. Macronutrients

The progressive increase in calcium content noted in shoots from 42 to 63 days after sowing (DAS) and in fruits from 7 to 21 days after anthesis (DAA) within landraces from KwaZulu-Natal and Limpopo (Table 2) could be the result of calcium immobility through the plants’ vascular tissues in maturing fruits [16]. Generally, mineral elements with low phloem mobility have significantly higher transportation rates from source (leaves and shoots) to target sink sites such as fruits and seeds [16]. Seed formation, which acts as a mineral element sink reservoir, also justifies the accumulation of calcium with fruit maturity and the resulting fruit hardening [17]. Calcium is necessary for skeletal health and reinforcement, neuromuscular function, blood clotting, blood pressure regulation, and overall immune system health [18]. 

The best harvest stages to obtain higher calcium content in shoots were 42 DAS for DSI and KSC, 49 DAS for KSC, and 56 DAS for KRI, as well as 63 DAS for BG-19, BG-26, BG-27, BG-100/GC, and ESC (Table 2). In fruits, the ideal harvest periods for high calcium levels were 7 DAA for DSI, ESC, KSC, KSP, NqSC, NSRC, and RSP; 14 DAA for KSC and RSP; and 21 DAA for NSRP. This variation can be attributed to the differences in the shoot and fruit growth rate, where some reach the maximum length and size earlier than others and thereby accumulated more calcium at their maximum growth stage [19]. Due to the immobility of calcium in the phloem tissue, it is more largely concentrated in the vegetative modules of plants than in fruits, which positively correlates the accumulation of macronutrients to their mobility in the phloem tissue rather than in the xylem tissue [20]. Furthermore, when fruit growth has ceased, the minerals are generally remobilized to other parts of the plant away from the fruit [21], which explains the lesser calcium content in the later harvest stage for other landraces. 

High levels of Mg can be obtained throughout all varying growth stages, with the landrace KRI recording the highest Mg levels at 42, 49 (along with KSC and NqSC), 56, and 63 DAS (Table 2). However, in fruits, only BG-27 at 21 DAA produced a significantly high Mg content. In the previous study, landrace KRI produced relatively smaller fruits compared to the majority of landraces investigated [13]. Cucurbit lines with small fruits have a shorter growth cycle than large-fruited ones [19]. This explains the superior nutrient status of KRI from early to mature stages and its hypothesized ability to accumulate its maximum mineral capacity sooner than other landraces. Again, the partitioning of Mg in shoots and fruits further emphasizes the significance of the mobility of minerals in the vascular tissues on the availability in different plant parts [22]. 

The investigated landraces had higher shoot (0.17–0.44 g/100 g) and fruit (0.074–2.532 g/100 g) magnesium content ranges (Table 2) than those of accessions from India (0.085 to 0.234 g/100 g) [23]. Although both studies were conducted during the summer months, Mg analysis was conducted only in tender juvenile fruits accessions from India [23] but at various stages of growth in the current study (Table 2). The higher Mg content in the current study could be attributed to a smaller genotypic pool of 19 landraces (Table 2) compared to the 96 accessions in India [23]. *Cucurbita maxima*, also from the Cucurbitaceae family and whose fruits change from green to yellow or orange upon maturity, recorded a lower Mg range of 0.056–1.978 g/100 g [24] than that of the current study. This high Mg content probably explains the maintenance of the green color (in different shades) in maturing *L. siceraria* fruits, because Mg is a component of the green chlorophyll pigment and is primarily concentrated in the pericarp of fruits, which decreases as the fruits mature from green to yellow and orange [15]. Magnesium is also involved in the synthesis of carotenoids known as carotenogenesis, which are the yellow and orange pigments (such as *ꞵ*-carotene) [25,26]. Additionally, Mg serves as a cofactor of enzymes in metabolic activity, protein synthesis, RNA and DNA synthesis, and the maintenance of nervous tissue and cell membranes in the human body [14].

The optimal harvest stages for high potassium levels in shoots were 42 and 49 DAS for landraces KRI, BG-19, and BG-26, whereas for phosphorus, the recommended harvest stages were 42, 56, and 63 DAS for KRI and 63 DAS for BG-19 (Table 2). The ideal harvest stages for higher potassium in fruits were 7 (DSI, NqRC, and RSP) and 14 DAA (RSP), whereas for phosphorus, it was 7 (BG-24, BG-100/GC, and KRI) and 14 DAA (KSC). The potassium content in both shoots and fruits did not vary across all stages of growth in the majority of landraces, though some variations in phosphorus content were observed, possibly due to *L. siceraria*’s ability to retain green-colored fruits from juvenile to mature [27]. The highest K and P content in landrace DSI, with solid, dark, green-colored fruits, and in landrace KSC, with solid, smooth pale-green fruits, as opposed to the lowest content in KRI, with green-colored fruits with cream warts, and NqSC, with solid, smooth, pale-green fruits across all growth phases (Table 2), may have resulted from differences in saturation of the green pigment chlorophyll in the fruit pericarp [21]. Lycopene, a carotenoid pigment, is responsible for the red flesh color of cucurbits like *Citrullus lanatus* [28]. This compound is positively correlated with K and P content in fruits, where they both increase as the fruit matures from green to red color [21]. Therefore, the low K and P content in *L. siceraria* fruits may be related to low lycopene content [21] because these fruits maintain different shades of green from juvenile to mature. Although *Momordica charantia* is a Cucurbitaceae family member that retains green fruits until maturity [21], it contained lower K (0.078–0.483 g/100 g) and P (0.02–0.08 g/100 g) contents ranges [5] compared with *L. siceraria* landraces in the current study (Table 3). These variations can be attributed to differences in genera (*Momordica* and *Lagenaria*) within the Cucurbitaceae family. 

Cucurbits are generally high in potassium, which is an important constituent of electrolytes and facilitates nerve impulses for optimal brain function and nervous system maintenance [24]. Moreover, potassium also assists as a vasodilator, relaxing the blood vessels and facilitating optimum oxygenation to vital organs, thus protecting against heart-related illnesses [5]. Phosphorus, on the other hand, is a necessary component of RNA, DNA, and nucleoproteins, which are responsible for cellular division, reproduction, and genetic transmission [29]. Phosphorus also helps with acid–base balance, kidney function, a regular heartbeat, and nerve impulses [18].

The nitrogen levels in shoots were notably higher (2.902–4.752 g/100 g) than those of fruits (0.59–3.39 g/100 g) in the current study (Table 2). This can be attributed to the function of vegetative modules of plants that manufacture energy, since nitrogen is a key component of the green pigment chlorophyll, capturing light energy and translocating assimilates to reservoir sink sites such as flowers and fruits [21,22]. A similar study on *Citrullus lanatus* grafted onto a *L. siceraria* rootstock recorded higher nitrogen content in shoots ranging from 4.164–5.991 g/100 g [30]. This can be attributed to the comparable study using younger and fresher shoots that were vigorously growing at 26 days after planting (DAP) [30], compared to the shoots at 42–63 DAS of the current study. 

All landraces consistently maintained a substantial and comparable carbon content throughout various growth stages (7–21 DAA) (Table 2). High carbon content is significant for preserving berry fruits and extending their shelf life because carbon inhibits ethylene synthesis and thus contributes to the maintenance of firm fruits [31]. Among the landraces examined in this study, KRI at 14 DAA exhibited the highest C/N ratio (Table 2). This elevated C/N ratio serves as an agronomic indicator associated with high-yielding plants [32]. A high C/N ratio is further positively associated with desirable traits, such as elevated total chlorophyll content, increased number of pistillate flowers, enhanced fruit retention, and ultimately, a higher yield per plant [32]. Moreover, it is negatively correlated with nitrogen [32]. This agrees with the current findings, where the landrace KRI produced the overall lowest nitrogen content despite having the overall highest C/N ratio at 14 DAA (Table 2). 

### 3.2. Micronutrients

All *L. siceraria* landraces in the current study had exceptionally low Na levels, ranging from 0.02 to 0.09 mg/kg for shoots and from 0.02 to 0.148 mg/kg for fruits (Table 3), which are significantly lower than the daily intake requirement of 2.3 g/day [33]. Though the Na content was low, the ideal harvest periods for shoots with high sodium content were 42 DAS for BG-19 and BG-100/GC; 56 DAS for BG-19, BG-24, BG-26, BG-80, DSI, ESC, KSC, KSP, and NqSC; and 63 DAS for BG-19, ESC, and KSP (Table 3). The best harvest for fruits with high sodium was 7 DAA for DSI and KSP, 14 DAA for BG-100/GC, and 21 DAA for BG-21, BG-27, KRI, NqSC, NSRC, and NSRP. The low Na levels make the shoots and fruits suitable for consumption by middle-aged to elderly people because low Na consumption helps prevent osteoporosis and hypertension [34]. The low Na content could be linked to the accumulation of sugars with fruit maturity, causing an increase in fruit osmotic pressure, leading to rind hardening, and resulting in an efflux of Na from fruits back to the vegetative modules [15]. This could possibly explain the fluctuating Na levels across all stages of shoot and fruit growth in the investigated landraces.

The harvest times of 42 DAS for BG-19 and BG-81, 49 DAS for BG-19, 56 DAS for BG-26, and 63 DAS for BG-19 were optimal for obtaining higher manganese content in shoots, and 7 DAA for BG-24, BG-100/GC, ESC, KRI, and NRC, and 14 DAA for KSC, NqRC, and NSRC in fruits (Table 3). Furthermore, for high iron levels in the shoots, landraces BG-100/GC and KSC could be harvested at 42–56 DAS and 63 DAS, respectively. For fruits, the ideal time for optimum iron was 7 DAA for BG-27, BG-31, BG-100/GC, ESC, and KRI, and 14 DAA for BG-24, DSI, KSC, KSP, NqRC, and NSRP. Moreover, for superior zinc content in shoots and fruits, the optimal harvest periods for BG-19, BG-26, BG-80, and ESC were 42 DAS; 49 DAS for BG-19, BG-24, BG-26, and BG-80; 56 DAS for BG-19 and BG-26; 63 DAS for BG-81; 7 DAA for BG-24, DSI, ESC, KRI; and 14 DAA for KSC and NqRC. In addition, the best harvest periods for high copper content were 42 DAS for BG-100/GC, BG-19, and KRI; 63 DAS for BG-19 in shoots; 7 DAA for BG-100/GC, ESC, KRI, and NRC; 14 DAA for KSC; and 21 DAA for DSI and NSRP for fruits (Table 3).

The shoot and fruit ranges for Mn (26–88 mg/kg and 1.50–66.05 mg/kg), Fe (116–6005 mg/kg and 6.00–2375.90 mg/kg), and Zn (53–126 mg/kg and 11.98–148.12 mg/kg) levels were higher (Table 3) compared to the genotypes from Egypt, with 16.2, 1000, and 46.5 mg/kg, respectively [3]. These differences were probably caused by the environmental variance, and the fruits from the current study were not separated from the seeds compared to the genotypes from Egypt, which only had fruit pulp (without seeds) analyzed for mineral composition. The results of the current study support that the developing seeds within fruits are strong sink sites for mineral elements, particularly those with low mobility in the vascular tissues, such as Mn, Fe, Zn, and Cu [21]. The low mobility may result in internal mobilization within the fruit and remobilization away from the fruit, facilitated by the vascular sap backflow via transpiration or the xylem tissue [21]. The apoplast of specialized cells transports mineral elements via the vascular tissues, specifically the xylem and transpiring vegetative tissues, where these mineral components are redirected and distributed by the phloem to tissues that do not contain xylem or transpire [16]. The mineral elements Fe, Zn, Mn, and Cu are some of the micronutrients that have low concentrations in fruits and a low translocation rate through the phloem [16]. However, the copper content of the genotypes from Egypt was higher (173 mg/kg) [3] than the 13 mg/kg recorded from the landraces investigated in the current study.

In general, the mineral composition of *L. siceraria* shoots and fruits is relatively high, indicating that they are good sources of dietary elements, with essential metabolic and medicinal properties [3]. Manganese is a component of many enzymes; it is necessary for appropriate brain function and healthy nervous system activity throughout the body, and it is essential for the proper and normal growth of human bone structure [18]. It is also beneficial for postmenopausal ladies and osteoporosis prevention [18]. Mn is also required for the creation of urea (arginase), the breakdown of free radicals (superoxide dismutase), the conversion of pyruvate to oxaloacetate (pyruvate carboxylase), the development of bone and connective tissue, as well as in the operation of the brain and pancreas [24]. Moreover, iron plays a key role in redox activities, strengthens the immune system, improves mental function, lessens sensations of exhaustion and lassitude, and is a large component of hemoglobin [24]. 

Zinc plays a crucial role in the structure of many enzymes required for the production of protein and genetic material, and also has a key role in taste perception, wound healing, sperm production, healthy immune system function, normal fetal development, normal growth, sexual maturation, and digestion [18]. Copper is a necessary element for the human body, modulating the activity of vitamins, hormones, enzymes, and respiratory pigments, and it also helps with iron absorption [35].

### 3.3. Proximate Analysis

Landraces BG-19, BG-24, BG-26, BG-27, BG-80, BG-81, and BG-100/GC from Limpopo, with low moisture content at 42 and 49 DAS (Table 4), probably indicates that their shoots had a high nutritional status [36]. The same can be said for fruits produced by landraces BG-27, BG-31, and DSI at 7–21 DAA and NSRP at 14 DAA. Therefore, landrace BG-27, with a notably low moisture content among and across all phases of growth in shoots and fruits, can be recommended for selection for superior nutrient status [36] and prolonged shelf life [24]. However, landraces KSC, NqSC, DSI, and KSP, with sappier shoots at 42 and 49 DAS, along with the watery fruits of BG-100/GC at 21 DAA, as well as landraces KRI, KSC, KSP, NqRC, NqSC, and NSRC at 7 DAA (Table 4), had high moisture content, which is a limiting factor to the technological suitability of plant products, making them prone to microorganism action and spoilage [24]. 

The general decrease in moisture content with fruit maturity from 7 and 14 DAA to 21 DAA (Table 4) can be associated with the increase in soluble solids as a result of the hydrolysis of starch into simple sugars such as glucose and fructose [26]. The production of these simple sugars generates an osmotic pull that attracts water to the fruit, which facilitates fruit growth and subsequently decreases the moisture content [37,38].

The acid detergent fiber (ADF) content varies among different cucurbit species with varying nutritional compositions [39]. Furthermore, the ADF content is indicative of digestibility, where fruits with low ADF have greater digestibility and nutrient availability than those with high ADF, resulting in lower consumption due to the lack of degradability of fruits [39]. Therefore, shoots of landraces BG-26 (18.40%), BG-27 (18.90%), and BG-80 (19.04%) at 42 DAS and BG-26 (19.61%) and BG-27 (18.59%) at 49 DAS, as well as fruits of BG-24 (11.43%), BG-31 (12.29%), ESC (12.50%), and RSP (12.62%) at 7 DAA and BG-100/GC (12.60%) and DSI (12.22%) at 14 DAA (Table 4), can be recommended for highly degradable fruits with readily available nutrients in large quantities compared to similar fruiting species, such as *Cucurbita maxima*, with 21.7%, and *Cucurbita argyrosperma*, with 40.4% ADF content [39].

The neutral detergent fiber (NDF) content found in the shoots at 42–49 DAS and fruits at 7 DAA (Table 4) was significantly lower (for shoots, 25.15–39.74% and for fruits, 17.08–29.04%) than that found in *Cucurbita pepo* (48.47% [40] and 53.09% [41]). However, as the shoots and fruits matured from 56 to 63 DAS and from 14 to 21 DAA, the NDF also increased to comparable levels, ranging from 30.71 to 59.06% in shoots and surpassing the NDF level of *C. pepo* at the range of 16.31–73.31% in fruits. In the early fruit development of *cucurbits* (*C. pepo*), the assimilates are directed and concentrated on growing fruits, where the simple sugars and starch levels are highly concentrated between 40 and 50 days after pollination (DAP) [42]. Therefore, the lower NDF levels recorded in the current study may be a result of highly concentrated simple sugars, which lower the ADF and NDF levels in related species such as *C. pepo* [41]. Furthermore, landraces BG-26 and BG-27 at 42–49 DAS, along with NRC at 7 DAA and KRI and NRC at 14 DAA, can be recommended for fruits with a high simple sugar content that is easily digestible [40].

The acid detergent lignin (ADL) content accounts for plant parts that are not digestible, where high amounts of ADL reduce the degradation of ADF, NDF, hemicellulose, and cellulose [43]. Hence, landraces BG-80 at 42 DAS, KSC at 49 and 63 DAS, and BG-100/GC at 56 DAS with low ADL can be selected for high digestibility and readily available nutrients, whereas the landrace KRI across all growth stages (42–63 DAS) had a notably high ADL content, suggesting that the landrace undergoes intense lignification, which results in slow water loss and prolonged shelf life, as observed in *C. maxima*, *C. pepo*, and *Cucurbita moschata* [44].

Optimum ash content was recorded in immature fruits of BG-100/GC, DSI, NqRC, and NqSC at 7 DAA ranging from 4.70 to 35.60 g/100 g (Table 4). This range was significantly higher than the ash average of *C. moschata* immature fruits at 1.018% [26]. Landraces with high ash content are indicative of high mineral content, which is essential for human growth and development [24]. The maturing fruits of the current investigation at 14 and 21 DAA further displayed a greater ash content range of 2.53–9.93 g/100 g, which was still higher than that of matured *C. moschata* fruits at 1.014% [26]. The decrease in ash content with the maturing fruits can be linked to high mineral absorption from the soil during early fruit development, which is utilized in various biological processes [26]. Furthermore, the dilution effect can also be the cause of the decrease in ash, as maturing fruits have an increased assimilate influx, leading to fruit elongation and expansion [26].

The highest fruit protein contents were recorded at 7 DAA for KRI, KSC, and NSRP and at 14 DAA for BG-100/GC, KSP, NqRC, and NSRC (Table 4). The protein content range of 5.38–21.15 g/100 g of *L. siceraria* landraces in the current study was greater than the protein content of *C. pepo* (8.2%), *C. melo* (6.9%), and *C. lanatus* (7.9%) in a similar study [45]. The higher protein content in the current study can be attributed to the presence of seeds since they are the primary reservoir sink sites for protein in Cucurbitaceae, whereas only the fruit pulp without seeds was assayed for mineral composition in *C. pepo*, *C. melo*, and *C. lanatus* [45]. Protein further assumes a vital role within the human body, constituting approximately 45% of its total composition [29]. Its indispensability lies in facilitating crucial physiological processes, including tissue repair, nutrient transport, and the construction and functional integrity of muscular tissues [29].

The optimal harvest periods for high carbohydrate content were 42 DAS for BG-27 and BG-26 and 49 DAS for ESC (Table 4), indicating that these landraces have a high caloric value for human nutrition [46]. Carbohydrates provide energy to facilitate cellular division and elongation during fruit growth by creating an osmotic pressure gradient, resulting in rapid initial shoot and fruit growth in cucurbits [42]. Carbohydrates are then stored as starch, which degrades to simple sugars (glucose and fructose) as the fruiting plants age [44]. As a result, the higher carbohydrate content at 42–49 DAS in comparison to that at 63 DAS across all landraces may be explained by the simple sugars being metabolized into energy and remodified cellular structures in aging fruits [42,44].

The milliequivalent ratios of K/Ca+Mg of the current study ranged from 1.95 to 3.08 g/100 g (Table 4), which was higher than the recommended level of 2.20 g/100 g and lower than the level recorded in *C. sativus* samples ranging from 3.91 to 4.48 g/100 g [47]. A ratio higher than 2.20 g/100 g renders humans—more specifically, men—to be prone to hypomagnesemia, which results from extremely low magnesium levels in the blood [47,48]. The shoots of landraces KRI, KSC, and KSP at 56 DAS and only KSC and KSP at 63 DAS contained the acceptable milliequivalent ratios of K/Ca+Mg; hence, at these harvest periods, their shoots can be recommended for consumption for optimum muscle and nerve function, blood sugar regulation, and lower risk of cardiovascular illnesses [47,48].

### 3.4. Correlation among Nutritional Components of Lagenaria siceraria Fruits 

The positive correlation between Ca and Mg, as well as that of Mg and K with P and N (Table 6), may be because these mineral elements share comparative chemical properties and, hence, compete for the sites of absorption, translocation, and function in plant tissues [49]. Therefore, the selection of any one of these mineral elements will lead to increased levels of other mineral elements [5] and thereby enhance the nutritional value of the investigated *L. siceraria* landraces. Similarly, a study on *Citrullus colocynthis* reported a strong positive correlation between K and Mg [4]. Furthermore, more similarities were reported where Ca and Mg were also associated positively with each other, possibly because both mineral elements are alkaline earth metals [4]. Correlation coefficient analysis evaluates the relationship between various characteristics by which cucurbit yield and nutritive value can be improved [50].

The negative correlation between fiber (ADF and NDF) and carbohydrates (Table 6) suggests that *L. siceraria* shoots with low ADF and NDF will have higher carbohydrates, which are readily metabolized into metabolic energy, thus having a higher caloric value for human consumption [42]. A similar study on *Cucurbita pepo* reported that high levels of carbohydrates lower the ADF and NDF content [41].

### 3.5. Principal Component and Cluster Analyses

The first five principal components with positive eigenvalues ≥ 1 and accounting for 90.218% in shoots and 89.918% in fruits of the total variance in this study (Table 7 and Table 8) were comparable to the first five principal components responsible for a total of 84.39% variability in *C. maxima*, *C. pepo*, *Cucurbita moschata*, and *Cucurbita ficifolia* [51]. This indicates that they contributed the most to the variation among the landraces observed in proximal variables [4,52]. Principal component analysis (PCA) was conducted to identify the main contributors to variability between investigated landraces with regards to their nutritional qualities [51].

The dominant variables, namely, Ca, Mg, K, P, N, Fe, ADL, carbohydrates, and protein, for the first principal component (PC1), which represented 42.312% of the total variance (Table 7), was similar to a positive association of Mg, K, and P with PC1, which contributed 65.1% of the total (79.7%) variance in *Momordica charantia* [6]. Again, the first principal component, with 39.12% variability, integrated Mg, K, P, and Fe as some of the dominant mineral elements responsible for variation in different cucurbits [51], as in the current study with different *L. siceraria* landraces. This confirms macronutrients as dominant variables responsible for variability with respect to the nutrient status of cucurbits [51], and they should be considered during crop advancement through selection [53].

The shoot loadings of different variables based on the first two principal components (59.906% variability) indicates that ADF, NDF, and moisture content, which were positively associated with landraces DSI, KSC, KSP, and NqSC, contributed a greater proportion to total variability, whereas Na, Mn, Zn, and carbohydrates associated with landraces BG-19, BG-24, BG-26, BG-27, and ESC contributed the least. On the other hand, the fruit loadings based on PC1 and PC2 suggest that ash, K, Ca, NDF, ADF, Mn, Na, fat, and Al were positively correlated with DSI, ESC, KSC, and NSRC, while Fe, Zn, protein, N, moisture content, C, Cu, P, and Mg were associated with BG-27. A similar association between C/N ratio and landraces NqSC, NSRC, and NqSC was recorded. This was validated by the proximity of landraces to the proximal components they associated with, as illustrated by the length of the vector lines in the biplot [52]. A similar study on *Cucumis melo* also reported the clustering of cultivars to the nutrient components they correlated with [52]. Therefore, while conducting selection and/or choosing parental germplasm for hybridization in *L. siceraria* for a superior nutritive value, these nutrient traits can be considered [4]. 

In shoots, landraces BG-100/GC and KRI, and in fruits, KSC and ESC, formed singletons in the dendrograms (Figure 4a,b), which indicates that they differ genetically from other landraces [53]. Cluster I comprised KRI, which was strongly associated with high K, Mg, and ADL levels across all growth stages (42–63 DAS), as well as Ca at 56 DAS and P at 42 DAS. Cluster II had BG-100/GC, which correlated with high Fe levels at 42, 49, and 56 DAS (Table 4). Furthermore, cluster III was the major cluster, grouping all other landraces, as represented in the biplot. Similarly, *Cucurbita maxima* accessions with similar nutrient content and availability clustered together [53].

## 4. Materials and Methods

### 4.1. Germplasm Sourcing and Field Layout

Nineteen landraces of *Lagenaria siceraria* from different agro-climatic regions in northern KwaZulu-Natal and Limpopo, South Africa, were investigated (Table 1). Landraces from KwaZulu-Natal were named according to their area of origin, represented by the first letter; fruit texture, represented by the second letter; and fruit shape, represented by the third letter (Table 1). Landraces from Limpopo were named by previous investigators based on their entry number and distinguished by their fruit and seed traits [54,55,56,57]. Seeds of the landraces were collected from Ga-Phasa (23.4057° S, 29.1557° E), Kgohloane (23.4739° S, 29.2213° E), Khangelani (29.0106° S, 31.2211° E), Moletjie-Mabokelele (23.4514° S, 29.1713° E), Emkhandlwini (28.508° E, 31.7002° E), Rorke’s Drift (28.3492° S, 30.5351° E), Nquthu (28.2195° S, 30.6746° E), and Dundee (28.1650° S, 30.2343° E). The field experiment was conducted during the summer seasons, September 2020–January 2021 and September 2021–January 2022. The experiment was conducted in the vegetable field unit of the Department of Botany, Faculty of Science, Agriculture and Engineering, University of Zululand, KwaDlangezwa campus (28.51° S, 31.50° E), which has a sub-tropical climate [58]. The KwaDlangezwa area has a daily mean temperature of 28.4 °C in summer and 14.5 °C in winter [59]. The study area receives an annual rainfall ranging from 299.95 to 350.02 mm [60].

The experiment adopted the randomized block design generated by R 4.2.1 software in the RStudio platform [61]. Seeds were directly sown into a 10 cm-deep pit with fertilizer NPK 2:3:4 (30) applied at planting at a rate of 400 kg/ha (40 g/m^2^ per pit) and placed below the seeds in 10–15 cm-deep pits. Experimental plots were 3 m × 4 m in size, and seeds were spaced at an intra-row spacing of 1 m and an inter-row spacing of 2 m. Each plot had 20 plants, with a net plot of 6 m^2^ having 6 plants. Each of the 19 landraces had 3 replicate plots, which resulted in 57 plots in total and bearing a total of 1140 plants. Weeding and insecticide applications were performed when necessary. The field was irrigated to field capacity for the duration of the experiment using a sprinkler system.

### 4.2. Sample Preparation 

For shoot nutrient composition, shoot tips with a minimum of three fully opened leaves were harvested at intervals of 42, 49, 56, and 63 days after sowing (DAS). Ten shoot tips were harvested from different plants within each plot. Therefore, shoots harvested from each plot constituted a replicate (*n* = 3). Fruits were harvested at intervals of 7, 14, and 21 days after anthesis (DAA). In each landrace, 10 fruits were harvested from different plants within each plot. Therefore, fruits from one plot constituted a replicate (*n* = 3). Harvested fruits were rinsed with tap water and cut into small pieces using a clean knife, and sun-dried for 24 h. The harvested shoots and fruits were rinsed with tap water and cut into small pieces using a clean knife, and sun-dried for 24 h. Thereafter, shoots and fruits were dried in an oven (Labcon incubator, Model 5016LC, LABCON, Krugersdorp, South Africa) at 65 °C until a constant dry mass was obtained. Shoots and fruits were ground into powder through a 0.84 mm sieve using a laboratory grinder (Hammer mill SMC, Scientific Manufacturing cc, Cape Town, South Africa) in preparation for analysis.

### 4.3. Target Proximate and Mineral Element Composition

*Lagenaria siceraria* landraces were assayed for moisture content, crude protein, and fiber in shoot tips and fruits, carbohydrates in shoots only, and ash, fats, and starch in fruits only, according to Association of Official Analytical Chemists (2000) methods [62]. Shoot tips and fruits were also analyzed for both macronutrients (nitrogen (N), phosphorus (P), calcium (Ca), potassium (K), and magnesium (Mg)) and micronutrients (sodium (Na), zinc (Zn), copper (Cu), manganese (Mn), and iron (Fe), but aluminum (Al) in fruits only) using the Soil Fertility and Analytical Services Laboratory [63]. Sub-samples of shoot tips and fruits were dried and ashed at 450 °C overnight. The ash was dissolved in 1 M HCl. The supernatant was analyzed for Al, Ca, Cu, K, Mg, Mn, Na, and Zn by atomic absorption spectroscopy (AAS). Nitrogen was determined before ashing using near-infrared reflectance [64]. Phosphorus concentrations were determined colorimetrically on a 2 mL aliquot of filtrate using a modification of the molybdenum blue procedure [63]. Potassium was determined from the extract directly on a flame photometer [65].

The following calibration and digestion protocols were observed for shoots and fruits of all landraces in triplicate samples. Inductively conducted plasma–optical emission spectrometry (ICP-OES) was conducted with the ICAP 7000 Thermo Fisher ICP-OES (Thermo Fisher Scientific, Waltham, MA, USA) in axial mode. It was calibrated to a radio frequency power of 1350 W, a nebulizer gas flow rate of 0.5 L min^−1^, a coolant gas flow rate of 12 L min^−1^, an auxiliary gas flow rate of 0.5 L min^−1^, and a pump speed of 35 revolutions per minute (rpm). The spectrometry reading was the mean of triplicate recordings of samples of each landrace, both for shoots and for fruits [66]. 

Approximately 1.0 g of the oven-dried shoot and fruit samples was placed in 100 mL volumetric flasks. Thereafter, 10 mL of nitric acid (HNO_3_) was added and kept overnight. On the following day, 8 mL of perchloric acid (HClO_4_) were added and gently swirled for 30 s. The flasks were then placed on the digestion unit at 100 °C and gradually increased to 260 °C until the production of red nitrogen dioxide (NO_2_) fumes stopped. The solution was further evaporated until the volume was reduced to about 3 to 5 mL but not to dryness. The end of the digestion was confirmed by the solution becoming colorless. The solution was allowed to cool down; then, 10 mL of distilled water were added to the volumetric flasks. Thereafter, the solution of each sample was filtered using Whatman No. 1 filter paper, and the aliquots were used for the determination of the desired mineral elements on the ICP-OES [67].

### 4.4. Nutrient Determination Methods

#### 4.4.1. Proteins 

Approximately 1.0 g of the sample was placed in a digestion flask; thereafter, 5 g of Kjeldahl catalyst were added along with 200 mL of concentrated sulfuric acid (H_2_SO_4_). A blank was created using a tube containing the chemicals without the sample. The flask was gently heated in an inclined position. As the bubbling ceased, it was boiled briskly until the solution cleared. Once the solution was cool, 60 mL of distilled water were gently added, and immediately thereafter the flask was connected to the digestion bulb on the condenser, which had the tip of the condenser immersed in standard acid and a receiver with 5–7 drops on the mixed indicator. The contents of the flask were mixed thoroughly by rotation of the flask and then heated until the ammonia (NH_3_) was distilled. The receiver was then removed, the tip of the condenser was washed, and titration was carried out for the excess standard acid distilled with standard sodium hydroxide (NaOH) solution. 

Thereafter, the protein content was calculated using the following formula described by AOAC 2000:Protein (%)=((A−B)×N×1.4007×6.25)÷W
where: *A* = volume (mL) of 0.2 N HCl used sample titration;*B* = volume (mL) of 0.2 N HCl used in blank titration;*N* = normality of HCl;*W* = weight (g) of sample;1.4007 = atomic weight of nitrogen;6.25 = the protein–nitrogen conversion factor for fish and its by-products.

#### 4.4.2. Ash 

The crucible and its lid were placed in a furnace set to 550 °C overnight as a form of sterilization. The crucible was then left for 30 min to cool in the desiccator. Thereafter, the crucible and its lid were weighed to 3 decimal places. Approximately 5 g of the sample were placed in the crucible with the lid half covered. It was then heated over a low Bunsen flame. When fumes were no longer produced, the crucible and lid were transferred to the furnace and left to heat at 550 °C overnight. During this step, the lid was not placed on the crucible. Once the heating was completed, the lid was placed to prevent the loss of fluffy ash, and then it was cooled in the desiccator. Once the ash turned grey, the crucible and lid were weighed. Thereafter, the ash content was calculated using the following formula documented by AOAC 2000:Ash (%)=(Weight of ash÷weight of sample)×100

#### 4.4.3. Fats 

To ensure the weight of the bottle and lid was stable before use, they were placed in an incubator at 105 °C overnight. Approximately 3–5 g of the sample were paper-filtered and wrapped. The sample was then placed in an extraction thimble and transferred to a Soxhlet extractor. Once the bottle was cool it was weighed, and thereafter, 250 mL of petroleum ether were poured into the bottle and placed on the heating mantle, which was switched on after the Soxhlet extraction apparatus was connected and water was turned on for cooling of the apparatus. The sample was heated for 14 h at a heat rate of 150 drops/min. The solvent was evaporated using the vacuum condenser. The bottle was then incubated at 80–90 °C to allow the solvent to completely evaporate, which then left the bottle completely dry. The bottle with a partially covered lid was transferred to the desiccator to cool. Once cool, the bottle and its dried content were reweighed. Thereafter, the fat content was calculated using the following formula described by AOAC 2000:$$\mathrm{Fat}\;(\%)=\left(Weight\;of\;fat÷Weight\;of\;sample\right)\times 100$$

### 4.5. Multivariate Statistical Analysis

Data were subjected to ANOVA using the GenStat 15th edition. Means were separated using Tukey’s honesty significant difference (HSD) test at the 5% significance level and used to construct interaction tables of variables at varying growth stages. Correlations and principal component analysis (PCA) were implemented to determine multi-character variation using GenStat 15th edition. Cluster analysis was performed on the genetic distance matrix by using the hierarchical cluster analysis (HCA) method through a biplot and dendrogram to study the relationship among landraces.

## 5. Conclusions

The investigated *Lagenaria siceraria* landraces had notable variations in their nutritional composition in sequentially harvested shoots (42–63 DAS) and fruits (7–21 DAA). Landraces KRI and DSI at 42–63 DAS produced shoots of higher nutritional value, while the fruits of ESC and KSC at 7–21 DAA were the most nutritious among all landraces. Most nutrient traits, much like calcium, exhibited a progressive increase with shoot and fruit maturity, emphasizing the significance of mineral element mobility on the availability of nutrients on various plant parts. Nutrient traits of similar chemical properties and function in plants correlated positively with each other, such as Ca and Mg, Mg with K, and K and N. The micronutrients (Ca, Mg, K, P, and N) in shoots were the main contributors to variability, whereas in fruits, macronutrients (Fe, Zn, Cu, and Al) contributed more to the variation based on the principal component analysis. The proximate composition also differed with maturity, where the majority of the landraces had lower ADF, NDF, and ADL, though they had high carbohydrates at the juvenile (42–49 DAS) stages compared to mature stages at 56–63 DAS. Hence, *L. siceraria* plant parts consumed early will be highly digestible and have a high caloric value. The biplot and dendrogram grouped landraces with similar nutrient attributes that also correlated positively with each other. The study has addressed the knowledge gap and recorded for the first time the variation in nutrient composition in sequentially harvested shoots and fruits of *L. siceraria* landraces across different growth stages, thus providing valuable insights into the nutritional dynamics of *L. siceraria* landraces across various growth stages and offering a basis for understanding and optimizing their most nutritive growth phase.

## Figures and Tables

**Figure 1 plants-13-01475-f001:**
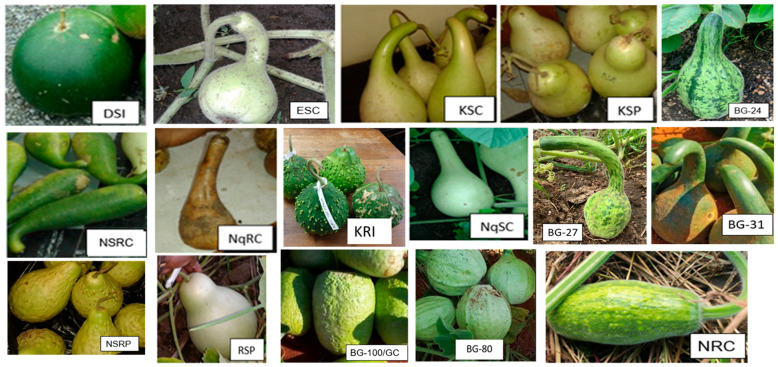
Fruit variation in *L. siceraria* landraces explained in Table 1.

**Figure 2 plants-13-01475-f002:**
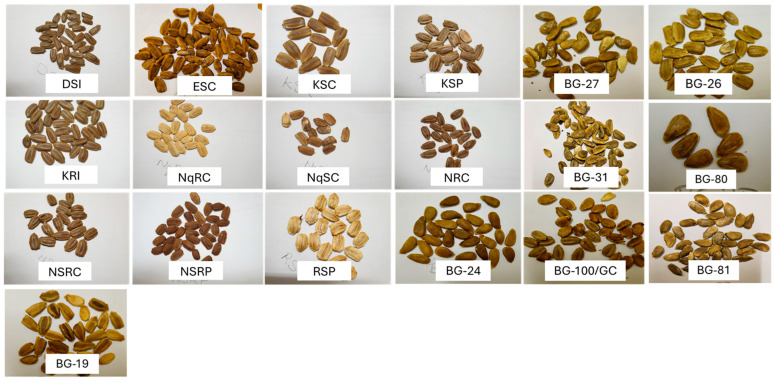
Seed variation in *L. siceraria* landraces explained in Table 1.

**Figure 3 plants-13-01475-f003:**
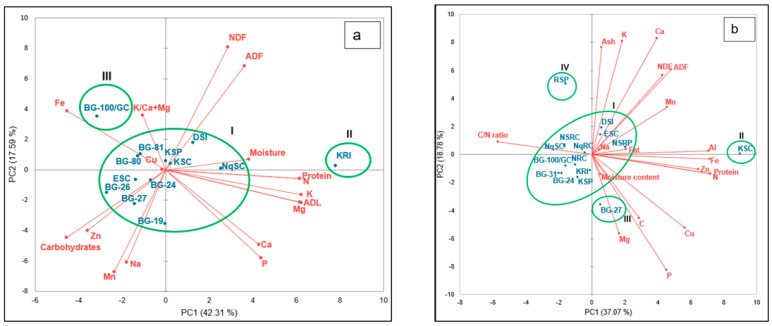
Biplot of *L. siceraria* landraces with nutritional components of shoots (**a**) and fruits (**b**). Landraces are explained in Table 1. Variables—Ca, calcium; Mg, magnesium; K, potassium; P, phosphorus; N, nitrogen; C, carbon; C/N, carbon/nitrogen ratio; Na, sodium; Mn, manganese; Fe, iron; Zn, zinc; Cu, copper; moisture, moisture content; ADF, acid detergent fiber; NDF, neutral detergent fiber; K/Ca+Mg, potassium/calcium+magnesium ratio; and ADL, acid detergent lignin.

**Figure 4 plants-13-01475-f004:**
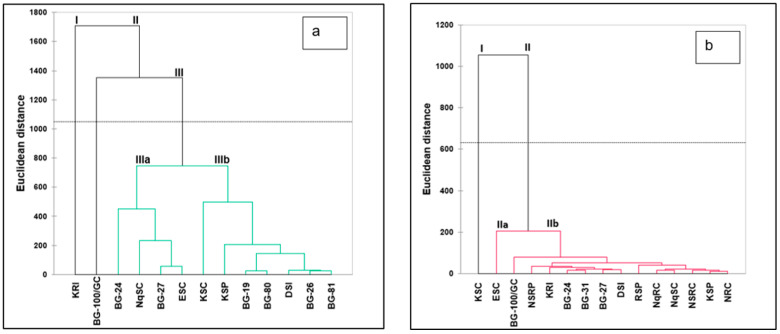
Hierarchical cluster showing dissimilarities amongst nutritional attributes of *L. siceraria* shoots (**a**) and fruits (**b**) at different growth stages using the complete linkage method. Landraces are explained in Table 1.

**Table 1 plants-13-01475-t001:** Description of landraces from northern KwaZulu-Natal and Limpopo [13].

Prov	LR	Area	Fruit Color	Fruit Texture	Fruit Shape	Seed Type	Seed Color	Seed Texture	Seed Size	Seed Line	Seed Shape
KZN	DSI	Dundee	Dark green	Smooth	Isodiametric	Siceraria	Dark brown	Smooth	Large	Present	Oblong
KZN	ESC	Emkhandlwini	Pale green	Smooth	Curvilinear	Asiatica	Light brown	Leathery	Medium	Present	Slightly oblong
KZN	KSC	Khangelani	Pale green	Smooth	Curvilinear	Asiatica	Brown	Leathery	Large	Present	Slightly oblong to rectangular
KZN	KSP	Khangelani	Pale green	Smooth	Pear	Asiatica	Brown	Leathery	Large	Present	Slightly oblong to rectangular
KZN	KRI	Khangelani	Green	Rough	Isodiametric	Siceraria	Dark brown	Leathery	Large	Present	Slightly oblong to rectangular
KZN	NqRC	Nquthu	Pale green	Rough	Curvilinear	Intermediate	Light brown	Leathery	Medium	Present	Slightly oblong
KZN	NqSC	Nquthu	Pale green	Smooth	Semi-curvilinear	Asiatica	Light brown	Leathery	Medium	Present	Slightly oblong
KZN	NRC	Ndumo	Dark green	Rough	Cylindrical	Siceraria	Creamy brown	Smooth	Small	Absent	Oblong
KZN	NSRC	Nquthu	Green	Semi-rough	Curvilinear	Intermediate	Brown	Leathery	Medium	Present	Slightly oblong
KZN	NSRP	Nquthu	Pale green	Semi-rough	Pear	Intermediate	Brown	Leathery	Medium	Present	Slightly oblong
KZN	RSP	Rorke’s Drift	Pale green	Smooth	Pear	Asiatica	Light brown	Leathery	Large	Present	Rectangular
LP	BG-19	Kgohloane	Dark green	Smooth	Isodiametric	Siceraria	Brown	Leathery	Large	Present	Slightly oblong to rectangular
LP	BG-24	Go-Phasa	Pale green	Corrugated	Cavate	Siceraria	Dark brown	Smooth	Small	Absent	Oblong
LP	BG-26	Kgohloane	Dark green	Smooth	Cavate	Intermediate	Brown	Leathery	Large	Present	Slightly oblong to rectangular
LP	BG-27	Kgohloane	Pale and dark green	Semi-rough	Cavate	Siceraria	Brown	Leathery	Small	Present	Oblong
LP	BG-31	Kgohloane	Dark green	Smooth	Cavate	Intermediate	Brown	Leathery	Large	Present	Slightly oblong to rectangular
LP	BG-80	Moletjie-Mabokelele	Pale green	Corrugated	Cavate	Asiatica	Dark brown	Smooth	Medium	Absent	Oblong
LP	BG-81	Kgohloane	Pale and dark green	Corrugated	Cavate	Asiatica	Brown	Leathery	Large	Present	Oblong
LP	BG-100/GC	Kgohloane	Pale green	Semi-rough	Cylindrical	Asiatica	Light brown	Leathery	Medium	Present	Slightly oblong

Prov—province, KZN—KwaZulu-Natal, LP—Limpopo, LR—landraces, BG—bottle gourd. The area of origin, fruit texture, and fruit shape were used to name the landraces from KZN. The naming of landraces from Limpopo was determined using the entry number based on the area of origin [13].

**Table 2 plants-13-01475-t002:** Variation in macronutrients of *Lagenaria siceraria* shoots and fruits harvested at different stages of growth.

Macronutrients	Plant Part	Landraces		Harvesting Period		
**Ca (g/100 g)**	**Shoots**		**42 DAS**	**49 DAS**	**56 DAS**	**63 DAS**
		BG-19	0.39 ^c–f^	0.33 ^e–k^	0.33 ^e–k^	0.50 ^a^
		BG-24	0.31 ^g–l^	0.38 ^c–g^	0.31 ^g–l^	0.33 ^e–k^
		BG-26	0.27 ^k–m^	0.36 ^c–i^	0.30 ^h–l^	0.40 ^c–e^
		BG-27	0.30 ^h–l^	0.33 ^e–k^	0.32 ^f–k^	0.41 ^b–d^
		BG-80	0.29 ^i–m^	0.29 ^i–m^	0.28 ^j–m^	0.31 ^g–l^
		BG-81	0.37 ^c–h^	0.40 ^c–e^	0.29 ^i–m^	0.41 ^b–d^
		BG-100/GC	0.22 ^m^	0.29 ^i–m^	0.24 ^lm^	0.31 ^g–l^
		DSI	0.40 ^c–e^	0.32 ^f–k^	0.35 ^d–j^	0.26 ^k–m^
		ESC	0.27 ^k–m^	0.37 ^c–h^	0.27 ^k–m^	0.38 ^c–g^
		KRI	0.37 ^c–h^	0.36 ^c–i^	0.48 ^ab^	0.40 ^c–e^
		KSC	0.41 ^b–d^	0.40 ^c–e^	0.35 ^d–j^	0.32 ^f–k^
		KSP	0.33 ^e–k^	0.43 ^a–c^	0.33 ^e–k^	0.39 ^c–f^
		NqSC	0.39 ^c–f^	0.38 ^c–g^	0.29 ^i–l^	0.38 ^c–g^
	**Fruits**		**7 DAA**	**14 DAA**	**21 DAA**	
		BG-24	0.082 ^mn^	0.136 ^g–n^	0.116 ^j–n^	
		BG-27	0.088 ^l–n^	0.108 ^j–n^	0.136 ^g–n^	
		BG-31	0.090 ^l–n^	0.099 ^k–n^	0.138 ^g–n^	
		BG-100/GC	0.102 ^k–n^	0.070 ^n^	0.202 ^d–g^	
		DSI	0.250 ^cd^	0.164 ^e–k^	0.158 ^f–l^	
		ESC	0.478 ^a^	0.119 ^i–n^	0.218 ^d–f^	
		KRI	0.136 ^g–n^	0.100 ^k–n^	0.142 ^g–m^	
		KSC	0.138 ^g–n^	0.434 ^a^	0.228 ^c–e^	
		KSP	0.212 ^d–f^	0.156 ^f–l^	0.100 ^k–n^	
		NqRC	0.142 ^g–m^	0.189 ^d–i^	0.199 ^d–g^	
		NqSC	0.214 ^d–f^	0.164 ^e–k^	0.152 ^f–l^	
		NRC	0.124 ^i–n^	0.126 ^h–n^	0.194 ^d–h^	
		NSRC	0.248 ^cd^	0.156 ^f–l^	0.176 ^e–j^	
		NSRP	0.138 ^g–n^	0.296 ^bc^	0.355 ^b^	
		RSP	0.296 ^bc^	0.334 ^b^	0.216 ^d–f^	
**Mg (g/100 g)**	**Shoots**		**42 DAS**	**49 DAS**	**56 DAS**	**63 DAS**
		BG-19	0.28 ^e–k^	0.25 ^g–l^	0.26 ^f–l^	0.33 ^b–f^
		BG-24	0.19 ^lm^	0.25 ^g–l^	0.25 ^g–l^	0.26 ^f–l^
		BG-26	0.22 ^j–m^	0.25 ^g–l^	0.26 ^f–l^	0.27 ^e–k^
		BG-27	0.22 ^j–m^	0.25 ^g–l^	0.25 ^g–l^	0.34 ^b–e^
		BG-80	0.21 ^k–m^	0.23 ^i–m^	0.26 ^f–l^	0.25 ^g–l^
		BG-81	0.22 ^j–m^	0.23 ^i–m^	0.22 ^j–m^	0.28 ^e–k^
		BG-100/GC	0.17 ^m^	0.22 ^j–m^	0.21 ^k–m^	0.24 ^h–m^
		DSI	0.31 ^d–h^	0.25 ^g–l^	0.26 ^f–l^	0.24 ^h–m^
		ESC	0.22 ^j–m^	0.26 ^f–l^	0.27 ^e–k^	0.33 ^b–f^
		KRI	0.39 ^a–c^	0.36 ^b–d^	0.44 ^a^	0.40 ^ab^
		KSC	0.32 ^c–g^	0.29 ^d–j^	0.31 ^d–h^	0.28 ^e–k^
		KSP	0.24 ^h–m^	0.27 ^e–k^	0.27 ^e–k^	0.28 ^e–k^
		NqSC	0.34 ^b–e^	0.30 ^d–i^	0.22 ^j–m^	0.27 ^e–k^
	**Fruits**		**7 DAA**	**14 DAA**	**21 DAA**	
		BG-24	0.168 ^b^	0.108 ^b^	0.166 ^b^	
		BG-27	0.156 ^b^	0.172 ^b^	2.532 ^a^	
		BG-31	0.142 ^b^	0.162 ^b^	0.188 ^b^	
		BG-100/GC	0.172 ^b^	0.116 ^b^	0.118 ^b^	
		DSI	0.246 ^b^	0.180 ^b^	0.177 ^b^	
		ESC	0.312 ^b^	0.144 ^b^	0.162 ^b^	
		KRI	0.244 ^b^	0.124 ^b^	0.124 ^b^	
		KSC	0.250 ^b^	0.378 ^b^	0.180 ^b^	
		KSP	0.228 ^b^	0.192 ^b^	0.152 ^b^	
		NqRC	0.078 ^b^	0.162 ^b^	0.170 ^b^	
		NqSC	0.078 ^b^	0.184 ^b^	0.182 ^b^	
		NRC	0.200 ^b^	0.166 ^b^	0.150 ^b^	
		NSRC	0.130 ^b^	0.200 ^b^	0.168 ^b^	
		NSRP	0.182 ^b^	0.220 ^b^	0.220 ^b^	
		RSP	0.144 ^b^	0.074 ^b^	0.132 ^b^	
**K (g/100 g)**	**Shoots**		**42 DAS**	**49 DAS**	**56 DAS**	**63 DAS**
		BG-19	4.01 ^a–c^	4.08 ^a–c^	3.55 ^a–c^	4.39 ^a–c^
		BG-24	3.09 ^bc^	3.55 ^a–c^	3.79 ^a–c^	3.70 ^a–c^
		BG-26	3.74 ^a–c^	4.13 ^a–c^	3.31 ^bc^	3.51 ^a–c^
		BG-27	3.22 ^bc^	3.61 ^a–c^	3.44 ^a–c^	4.08 ^a–c^
		BG-80	3.30 ^bc^	3.42 ^a–c^	4.09 ^a–c^	3.35 ^a–c^
		BG-81	2.98 ^bc^	3.50 ^a–c^	3.20 ^bc^	4.03 ^a–c^
		BG-100/GC	2.78 ^c^	3.74 ^a–c^	3.52 ^a–c^	3.64 ^a–c^
		DSI	4.62 ^a–c^	4.30 ^a–c^	3.68 ^a–c^	3.32 ^a–c^
		ESC	3.31 ^bc^	3.90 ^a–c^	4.22 ^a–c^	4.52 ^a–c^
		KRI	5.68 ^a^	4.53 ^a–c^	4.59 ^a–c^	4.73 ^a–c^
		KSC	4.84 ^a–c^	3.93 ^a–c^	3.69 ^a–c^	2.97 ^bc^
		KSP	3.77 ^a–c^	4.03 ^a–c^	3.23 ^bc^	3.31 ^bc^
		NqSC	5.28 ^ab^	5.05 ^a–c^	3.61 ^a–c^	3.93 ^a–c^
	**Fruits**		**7 DAA**	**14 DAA**	**21 DAA**	
		BG-24	2.81 ^c–e^	2.42 ^c–e^	2.70 ^c–e^	
		BG-27	2.34 ^de^	2.89 ^c–e^	3.01 ^c–e^	
		BG-31	2.22 ^de^	2.25 ^de^	2.81 ^c–e^	
		BG-100/GC	3.55 ^b–e^	2.02 ^e^	2.67 ^c–e^	
		DSI	8.91 ^a^	2.33 ^de^	2.82 ^c–e^	
		ESC	2.97 ^c–e^	2.46 ^c–e^	3.61 ^b–e^	
		KRI	3.61 ^b–e^	1.62 ^e^	1.93 ^e^	
		KSC	3.02 ^c–e^	3.65 ^b–e^	3.12 ^c–e^	
		KSP	3.04 ^c–e^	2.74 ^c–e^	1.76 ^e^	
		NqRC	4.42 ^b–d^	2.33 ^de^	1.71 ^e^	
		NqSC	3.64 ^b–e^	1.87 ^e^	2.47 ^c–e^	
		NRC	2.02 ^e^	1.68 ^e^	1.86 ^e^	
		NSRC	2.58 ^c–e^	2.51 ^c–e^	2.10 ^de^	
		NSRP	2.59 ^c–e^	3.06 ^c–e^	3.29 ^c–e^	
		RSP	4.69 ^bc^	5.85 ^b^	2.88 ^c–e^	
**P (g/100 g)**	**Shoots**		**42 DAS**	**49 DAS**	**56 DAS**	**63 DAS**
		BG-19	0.64 ^b–d^	0.58 ^d–g^	0.63 ^b–e^	0.70 ^b^
		BG-24	0.38 ^q–t^	0.47 ^j–p^	0.45 ^l–q^	0.56 ^e–h^
		BG-26	0.63 ^b–e^	0.65 ^b–d^	0.53 ^g–k^	0.50 ^h–n^
		BG-27	0.43 ^n–s^	0.49 ^h–o^	0.42 ^o–s^	0.61 ^c–f^
		BG-80	0.43 ^n–s^	0.42 ^o–s^	0.52 ^g–l^	0.49 ^h–o^
		BG-81	0.36 ^st^	0.38 ^q–t^	0.38 ^q–t^	0.52 ^g–l^
		BG-100/GC	0.33 ^t^	0.43 ^n–s^	0.42 ^o–s^	0.47 ^j–p^
		DSI	0.45 ^l–q^	0.55 ^f–i^	0.48 ^i–o^	0.54 ^f–j^
		ESC	0.45 ^l–q^	0.44 ^m–r^	0.54 ^f–j^	0.55 ^f–j^
		KRI	0.81 ^a^	0.61 ^c–f^	0.68 ^bc^	0.70 ^b^
		KSC	0.55 ^f–i^	0.51 ^g–m^	0.48 ^i–o^	0.43 ^n–s^
		KSP	0.49 ^h–o^	0.48 ^i–o^	0.42 ^o–s^	0.37 ^r–t^
		NqSC	0.46 ^k–p^	0.40 ^p–t^	0.47 ^j–p^	0.51 ^g–m^
	**Fruits**		**7 DAA**	**14 DAA**	**21 DAA**	
		BG-24	0.466 ^b–g^	0.312 ^l–r^	0.457 ^b–h^	
		BG-27	0.474 ^b–f^	0.515 ^bc^	0.485 ^b–e^	
		BG-31	0.348 ^h–p^	0.405 ^c–l^	0.411 ^c–l^	
		BG-100/GC	0.481 ^b–e^	0.364 ^f–o^	0.142 ^t–w^	
		DSI	0.112 ^vw^	0.377 ^e–m^	0.455 ^b–i^	
		ESC	0.505 ^b–d^	0.399 ^c–m^	0.178 ^s–w^	
		KRI	0.567 ^b^	0.284 ^m–s^	0.308 ^l–r^	
		KSC	0.508 ^b–d^	0.763 ^a^	0.254 ^o–t^	
		KSP	0.432 ^c–j^	0.356 ^g–p^	0.314 ^k–r^	
		NqRC	0.199 ^r–w^	0.329 ^j–q^	0.246 ^p–u^	
		NqSC	0.100 ^w^	0.285 ^m–s^	0.460 ^b–h^	
		NRC	0.397 ^d–m^	0.372 ^e–n^	0.258 ^n–t^	
		NSRC	0.202 ^r–w^	0.429 ^c–k^	0.228 ^q–v^	
		NSRP	0.432 ^c–j^	0.408 ^c–l^	0.339 ^i–q^	
		RSP	0.178 ^s–w^	0.118 ^vw^	0.134 ^u–w^	
**N (g/100 g)**	**Shoots**		**42 DAS**	**49 DAS**	**56 DAS**	**63 DAS**
		BG-19	3.552 ^a^	3.278 ^a^	3.877 ^a^	3.499 ^a^
		BG-24	2.963 ^a^	3.147 ^a^	3.322 ^a^	4.264 ^a^
		BG-26	3.973 ^a^	3.950 ^a^	3.643 ^a^	3.258 ^a^
		BG-27	3.264 ^a^	2.840 ^a^	3.552 ^a^	3.090 ^a^
		BG-80	3.406 ^a^	3.274 ^a^	3.912 ^a^	3.714 ^a^
		BG-81	3.059 ^a^	3.381 ^a^	3.304 ^a^	3.766 ^a^
		BG-100/GC	2.902 ^a^	3.018 ^a^	3.318 ^a^	3.606 ^a^
		DSI	4.080 ^a^	3.907 ^a^	3.432 ^a^	3.854 ^a^
		ESC	3.688 ^a^	3.304 ^a^	3.590 ^a^	3.400 ^a^
		KRI	4.752 ^a^	3.510 ^a^	4.357 ^a^	4.206 ^a^
		KSC	3.648 ^a^	3.574 ^a^	3.059 ^a^	3.805 ^a^
		KSP	3.608 ^a^	4.147 ^a^	3.651 ^a^	2.990 ^a^
		NqSC	4.214 ^a^	4.280 ^a^	3.277 ^a^	3.958 ^a^
	**Fruits**		**7 DAA**	**14 DAA**	**21 DAA**	
		BG-24	1.70 ^b–f^	0.98 ^c–f^	1.23 ^b–f^	
		BG-27	1.55 ^b–f^	1.40 ^b–f^	1.40 ^b–f^	
		BG-31	1.05 ^b–f^	0.98 ^c–f^	1.03 ^b–f^	
		BG-100/GC	1.94 ^b–e^	1.26 ^b–f^	0.81 ^ef^	
		DSI	1.16 ^b–f^	1.88 ^b–e^	1.26 ^b–f^	
		ESC	1.49 ^b–f^	0.97 ^c–f^	0.88 ^c–f^	
		KRI	2.03 ^bc^	0.59 ^f^	0.81 ^ef^	
		KSC	1.95 ^b–e^	3.39 ^a^	1.53 ^b–f^	
		KSP	1.72 ^b–f^	1.86 ^b–e^	0.98 ^c–f^	
		NqRC	1.17 ^b–f^	1.83 ^b–e^	1.03 ^b–f^	
		NqSC	1.15 ^b–f^	1.34 ^b–f^	1.09 ^b–f^	
		NRC	1.31 ^b–f^	1.12 ^b–f^	0.86 ^d–f^	
		NSRC	1.06 ^b–f^	1.98 ^b–d^	1.16 ^b–f^	
		NSRP	2.17 ^b^	1.28 ^b–f^	1.19 ^b–f^	
		RSP	1.21 ^b–f^	1.07 ^b–f^	0.89 ^c–f^	
**C (g/100 g)**	**Fruits**		**7 DAA**	**14 DAA**	**21 DAA**	
		BG-24	38.37 ^a^	38.84 ^a^	39.17 ^a^	
		BG-27	40.24 ^a^	40.25 ^a^	39.39 ^a^	
		BG-31	36.78 ^a^	38.50 ^a^	38.89 ^a^	
		BG-100/GC	37.53 ^a^	38.73 ^a^	35.64 ^a^	
		DSI	33.70 ^a^	38.24 ^a^	39.94 ^a^	
		ESC	35.99 ^a^	36.90 ^a^	41.10 ^a^	
		KRI	37.10 ^a^	37.20 ^a^	38.90 ^a^	
		KSC	37.99 ^a^	35.35 ^a^	42.66 ^a^	
		KSP	36.90 ^a^	37.70 ^a^	38.00 ^a^	
		NqRC	36.09 ^a^	39.39 ^a^	37.60 ^a^	
		NqSC	37.13 ^a^	38.86 ^a^	39.90 ^a^	
		NRC	37.20 ^a^	37.30 ^a^	38.30 ^a^	
		NSRC	35.80 ^a^	37.60 ^a^	39.50 ^a^	
		NSRP	37.80 ^a^	42.00 ^a^	41.50 ^a^	
		RSP	35.10 ^a^	34.80 ^a^	41.40 ^a^	
**C/N ratio** **(g/100 g)**	**Fruits**		**7 DAA**	**14 DAA**	**21 DAA**	
		BG-24	22.63 ^k–p^	42.31 ^b–g^	31.97 ^g–m^	
		BG-27	26.02 ^i–p^	28.66 ^h–p^	28.20 ^h–p^	
		BG-31	35.08 ^c–j^	39.36 ^b–h^	37.69 ^b–h^	
		BG-100/GC	19.30 ^o–q^	30.84 ^g–o^	44.28 ^b–f^	
		DSI	29.05 ^h–p^	20.39 ^m–q^	31.63 ^g–n^	
		ESC	24.18 ^j–p^	38.00 ^b–h^	46.60 ^bc^	
		KRI	18.28 ^pq^	62.63 ^a^	48.26 ^b^	
		KSC	19.51 ^o–q^	10.44 ^q^	27.98 ^h–p^	
		KSP	21.45 ^l–q^	20.27 ^n–q^	38.97 ^b–h^	
		NqRC	30.81 ^g–o^	21.54 ^l–q^	36.50 ^c–i^	
		NqSC	32.17 ^g–l^	28.96 ^h–p^	36.61 ^b–i^	
		NRC	28.40 ^h–p^	33.30 ^e–k^	44.53 ^b–e^	
		NSRC	33.77 ^e–k^	18.99 ^pq^	34.05 ^e–k^	
		NSRP	17.42 ^pq^	32.81 ^f–l^	34.80 ^d–j^	
		RSP	29.01 ^h–p^	32.52 ^g–l^	46.36 ^b–d^	
**K/Ca+Mg (g/100 g)**	**Shoots**		**42 DAS**	**49 DAS**	**56 DAS**	**63 DAS**
		BG-19	2.64 ^a^	2.82 ^a^	2.40 ^a^	2.15 ^a^
		BG-24	2.54 ^a^	2.30 ^a^	2.69 ^a^	2.50 ^a^
		BG-26	3.03 ^a^	2.74 ^a^	2.33 ^a^	2.13 ^a^
		BG-27	2.49 ^a^	2.49 ^a^	2.41 ^a^	2.15 ^a^
		BG-80	2.66 ^a^	2.62 ^a^	2.96 ^a^	2.38 ^a^
		BG-81	2.08 ^a^	2.30 ^a^	2.51 ^a^	2.37 ^a^
		BG-100/GC	2.85 ^a^	2.94 ^a^	3.08 ^a^	2.64 ^a^
		DSI	2.60 ^a^	3.01 ^a^	2.42 ^a^	2.95 ^a^
		ESC	2.68 ^a^	2.50 ^a^	3.02 ^a^	2.51 ^a^
		KRI	2.87 ^a^	2.43 ^a^	1.95 ^a^	2.42 ^a^
		KSC	2.65 ^a^	2.29 ^a^	2.20 ^a^	1.95 ^a^
		KSP	2.66 ^a^	2.36 ^a^	2.14 ^a^	1.99 ^a^
		NqSC	2.85 ^a^	2.83 ^a^	2.83 ^a^	2.44 ^a^

Means followed by different superscript letters show significant differences in traits within and across growth stages (columns) for each landrace. The letters indicate the level of significance among landraces during various growth phases. This is based on Tukey’s honest significant difference test, with a significance level of *p* < 0.05. Landraces are explained in Table 1. Mac—macronutrients, DAS—days after sowing, DAA—days after anthesis, Ca—calcium, Mg—magnesium, K—potassium, P—phosphorus, N—nitrogen, C—carbon, C/N ratio—carbon/nitrogen ratio, and K/Ca+Mg—potassium/calcium+magnesium ratio in g/100 g.

**Table 3 plants-13-01475-t003:** Variation in micronutrients of *Lagenaria siceraria* shoots and fruits harvested at different stages of growth.

Micronutrients	Plant Part	Landraces		Harvesting Period		
**Na (mg/kg)**	**Shoots**		**42 DAS**	**49 DAS**	**56 DAS**	**63 DAS**
		BG-19	0.06 ^b–d^	0.02 ^f^	0.09 ^a^	0.07 ^a–c^
		BG-24	0.04 ^d–f^	0.03 ^ef^	0.08 ^ab^	0.06 ^b–d^
		BG-26	0.04 ^d–f^	0.04 ^d–f^	0.09 ^a^	0.05 ^c–e^
		BG-27	0.03 ^ef^	0.04 ^d–f^	0.06 ^b–d^	0.06 ^b–d^
		BG-80	0.03 ^ef^	0.03 ^ef^	0.08 ^ab^	0.05 ^c–e^
		BG-81	0.04 ^d–f^	0.02 ^f^	0.06 ^b–d^	0.05 ^c–e^
		BG-100/GC	0.06 ^b–d^	0.02 ^f^	0.05 ^c–e^	0.05 ^c–e^
		DSI	0.04 ^d–f^	0.02 ^f^	0.09 ^a^	0.04 ^d–f^
		ESC	0.02 ^f^	0.04 ^d–f^	0.09 ^a^	0.07 ^a–c^
		KRI	0.02 ^f^	0.05 ^c–e^	0.04 ^d–f^	0.04 ^d–f^
		KSC	0.05 ^c–e^	0.04 ^d–f^	0.07 ^a–c^	0.04 ^d–f^
		KSP	0.04 ^d–f^	0.02 ^f^	0.09 ^a^	0.07 ^a–c^
		NqSC	0.05 ^c–e^	0.05 ^c–e^	0.09 ^a^	0.04 ^d–f^
	**Fruits**		**7 DAA**	**14 DAA**	**21 DAA**	
		BG-100/GC	0.026 ^st^	0.064 ^i–n^	0.014 ^t^	
		BG-24	0.054 ^j–q^	0.066 ^i–m^	0.096 ^c–f^	
		BG-27	0.072 ^g–l^	0.084 ^e–i^	0.094 ^c–g^	
		BG-31	0.049 ^l–r^	0.102 ^c–e^	0.108 ^b–d^	
		DSI	0.138 ^a^	0.076 ^f–j^	0.086 ^d–i^	
		ESC	0.089 ^d–h^	0.096 ^c–f^	0.114 ^bc^	
		KRI	0.042 ^n–s^	0.022 ^st^	0.148 ^a^	
		KSC	0.024 ^st^	0.052 ^k–q^	0.066 ^i–m^	
		KSP	0.060 ^j–o^	0.022 ^st^	0.044 ^m–s^	
		NqRC	0.032 ^q–t^	0.028 ^r–t^	0.036 ^p–t^	
		NqSC	0.036 ^p–t^	0.040 ^o–s^	0.130 ^ab^	
		NRC	0.028 ^r–t^	0.032 ^q–t^	0.036 ^p–t^	
		NSRC	0.026 ^st^	0.034 ^q–t^	0.068 ^h–l^	
		NSRP	0.026 ^st^	0.074 ^f–k^	0.094 ^c–g^	
		RSP	0.026 ^st^	0.058 ^j–p^	0.054 ^j–q^	
**Mn (mg/kg)**	**Shoots**		**42 DAS**	**49 DAS**	**56 DAS**	**63 DAS**
		BG-19	65 ^a–e^	56 ^b–g^	70 ^ab^	70 ^ab^
		BG-24	37 ^f–k^	44 ^d–k^	46 ^c–k^	51 ^b–j^
		BG-26	40 ^f–k^	34 ^g–k^	88 ^a^	67 ^a–d^
		BG-27	33 ^g–k^	35 ^f–k^	39 ^f–k^	48 ^b–k^
		BG-80	33 ^g–k^	38 ^f–k^	41 ^f–k^	53 ^b–i^
		BG-81	42 ^e–k^	42 ^e–k^	43 ^e–k^	54 ^b–h^
		BG-100/GC	27 ^k^	40 ^f–k^	48 ^b–k^	52 ^b–i^
		DSI	28 ^jk^	32 ^h–k^	36 ^f–k^	41 ^f–k^
		ESC	30 ^i–k^	37 ^f–k^	38 ^f–k^	46 ^c–k^
		KRI	30 ^i–k^	26 ^k^	37 ^f–k^	31 ^h–k^
		KSC	30 ^i–k^	31 ^h–k^	51 ^b–j^	68 ^a–c^
		KSP	31 ^h–k^	41 ^f–k^	44 ^d–k^	46 ^c–k^
		NqSC	38 ^f–k^	38 ^f–k^	42 ^e–k^	58 ^b–f^
	**Fruits**		**7 DAA**	**14 DAA**	**21 DAA**	
		BG-24	20.01 ^cd^	4.00 ^hi^	10.02 ^f–h^	
		BG-27	1.50 ^i^	2.50 ^i^	1.50 ^i^	
		BG-31	17.99 ^c–e^	13.97 ^d–g^	18.03 ^c–e^	
		BG-100/GC	22.05 ^c^	12.00 ^e–g^	10.00 ^f–h^	
		DSI	9.99 ^f–h^	17.97 ^c–e^	7.98 ^g–i^	
		ESC	33.21 ^b^	13.99 ^d–g^	15.98 ^c–f^	
		KRI	21.98 ^c^	8.00 ^g–i^	12.00 ^e–g^	
		KSC	14.00 ^d–g^	66.05 ^a^	14.00 ^d–g^	
		KSP	22.02 ^c^	20.01 ^cd^	14.01 ^d–g^	
		NqRC	9.99 ^f–h^	22.05 ^c^	13.97 ^d–g^	
		NqSC	10.01 ^f–h^	14.03 ^d–g^	14.01 ^d–g^	
		NRC	16.03 ^c–f^	14.00 ^d–g^	8.01 ^g–i^	
		NSRC	19.97 ^cd^	20.04 ^cd^	15.97 ^c–f^	
		NSRP	8.00 ^g–i^	14.00 ^d–g^	12.02 ^e–g^	
		RSP	14.00 ^d–g^	18.00 ^c–e^	14.01 ^d–g^	
**Fe (mg/kg)**	**Shoots**		**42 DAS**	**49 DAS**	**56 DAS**	**63 DAS**
		BG-19	2433 ^m^	2023 ^o^	2456 ^m^	2820 ^k^
		BG-24	1062 ^vw^	1257 ^r–t^	1214 ^s–u^	2604 ^l^
		BG-26	1184 ^tu^	506 ^A^	4848 ^d^	3756 ^gh^
		BG-27	1873 ^p^	882 ^xy^	3468 ^ij^	2138 ^o^
		BG-80	1592 ^q^	2857 ^k^	1339 ^r^	3903 ^f^
		BG-81	3579 ^i^	1117 ^uv^	3481 ^ij^	2041 ^o^
		BG-100/GC	3879 ^f^	2452 ^m^	5046 ^c^	3422 ^j^
		DSI	590 ^A^	1308 ^rs^	3542 ^i^	4889 ^d^
		ESC	1774 ^p^	2031 ^o^	2278 ^n^	2055 ^o^
		KRI	271 ^B^	116 ^C^	307 ^B^	2310 ^n^
		KSC	730 ^z^	1238 ^r–t^	4201 ^e^	6005 ^a^
		KSP	966 ^wx^	846 ^yz^	3719 ^h^	5350 ^b^
		NqSC	478 ^A^	475 ^A^	2508 ^lm^	3856 ^fg^
	**Fruits**		**7 DAA**	**14 DAA**	**21 DAA**	
		BG-24	94.10 ^gh^	122.00 ^e^	46.10 ^no^	
		BG-27	100.00 ^fg^	55.90 ^l^	54.10 ^lm^	
		BG-31	133.90 ^d^	47.90 ^mn^	38.10 ^p–r^	
		BG-100/GC	232.50 ^c^	104.00 ^f^	18.00 ^u^	
		DSI	35.90 ^p–r^	137.80 ^d^	45.90 ^no^	
		ESC	411.80 ^b^	91.90 ^hi^	20.00 ^tu^	
		KRI	133.90 ^d^	34.00 ^qr^	84.00 ^j^	
		KSC	56.00 ^l^	2375.90 ^a^	18.00 ^u^	
		KSP	48.00 ^mn^	70.00 ^k^	42.00 ^n–p^	
		NqRC	6.00 ^v^	82.20 ^j^	31.90 ^rs^	
		NqSC	20.00 ^tu^	24.00 ^tu^	40.00 ^o–q^	
		NRC	40.10 ^o–q^	40.00 ^o–q^	86.10 ^ij^	
		NSRC	20.00 ^tu^	74.10 ^k^	39.90 ^o–q^	
		NSRP	40.00 ^o–q^	102.00 ^f^	54.10 ^lm^	
		RSP	18.00 ^u^	4.00 ^v^	26.00 ^st^	
**Zn (mg/kg)**	**Shoots**		**42 DAS**	**49 DAS**	**56 DAS**	**63 DAS**
		BG-19	104 ^a–e^	89 ^c–j^	105 ^a–e^	118 ^ab^
		BG-24	76 ^h–o^	84 ^d–j^	89 ^c–j^	104 ^a–e^
		BG-26	92 ^c–i^	90 ^c–j^	105 ^a–e^	111 ^a–c^
		BG-27	68 ^j–o^	73 ^i–o^	78 ^f–n^	101 ^b–f^
		BG-80	85 ^d–j^	89 ^c–j^	98 ^b–h^	126 ^a^
		BG-81	76 ^h–o^	79 ^f–n^	85 ^d–j^	104 ^a–e^
		BG-100/GC	71 ^i–o^	83 ^e–k^	98 ^b–h^	107 ^a–d^
		DSI	56 ^no^	77 ^g–n^	83 ^e–k^	121 ^ab^
		ESC	84 ^d–j^	82 ^e–l^	93 ^c–i^	93 ^c–i^
		KRI	76 ^h–o^	60 ^k–o^	82 ^e–l^	73 ^i–o^
		KSC	53 ^o^	80 ^f–m^	98 ^b–h^	100 ^b–g^
		KSP	71 ^i–o^	76 ^h–o^	91 ^c–j^	100 ^b–g^
		NqSC	58 ^m–o^	59 ^l–o^	87 ^d–j^	117 ^ab^
	**Fruits**		**7 DAA**	**14 DAA**	**21 DAA**	
		BG-24	50.03 ^c^	33.99 ^f–j^	40.06 ^c–g^	
		BG-27	42.01 ^c–f^	39.94 ^c–g^	40.06 ^c–g^	
		BG-31	37.98 ^d–h^	35.94 ^e–i^	38.05 ^d–h^	
		BG-100/GC	48.11 ^cd^	39.98 ^c–g^	22.00 ^k–n^	
		DSI	67.90 ^b^	49.92 ^c^	35.91 ^e–i^	
		ESC	69.74 ^b^	35.97 ^e–i^	19.97 ^l–n^	
		KRI	45.95 ^c–e^	22.00 ^k–n^	27.99 ^h–m^	
		KSC	39.99 ^c–g^	148.12 ^a^	29.99 ^g–l^	
		KSP	36.04 ^e–i^	36.01 ^e–i^	26.01 ^i–m^	
		NqRC	11.98 ^n^	40.10 ^c–g^	23.95 ^j–m^	
		NqSC	18.03 ^mn^	30.05 ^g–l^	40.02 ^c–g^	
		NRC	36.06 ^e–i^	31.99 ^f–k^	26.03 ^i–m^	
		NSRC	49.93 ^c^	48.09 ^cd^	19.96 ^l–n^	
		NSRP	31.99 ^f–k^	22.00 ^k–n^	30.05 ^g–l^	
		RSP	30.01 ^g–l^	29.99 ^g–l^	24.01 ^j–m^	
**Cu (mg/kg)**	**Shoots**		**42 DAS**	**49 DAS**	**56 DAS**	**63 DAS**
		BG-19	15 ^a–e^	11 ^c–e^	14 ^a–e^	21 ^a^
		BG-24	10 ^c–e^	15 ^a–e^	15 ^a–e^	20 ^ab^
		BG-26	12 ^c–e^	12 ^c–e^	16 ^a–d^	12 ^c–e^
		BG-27	8 ^e^	9 ^de^	11 ^c–e^	17 ^a–c^
		BG-80	10 ^c–e^	11 ^c–e^	13 ^b–e^	14 ^a–e^
		BG-81	10 ^c–e^	11 ^c–e^	12 ^c–e^	16 ^a–d^
		BG-100/GC	20 ^ab^	14 ^a–e^	13 ^b–e^	15 ^a–e^
		DSI	10 ^c–e^	13 ^b–e^	14 ^a–e^	16 ^a–d^
		ESC	10 ^c–e^	11 ^c–e^	14 ^a–e^	16 ^a–d^
		KRI	15 ^a–e^	10 ^c–e^	17 ^a–c^	14 ^a–e^
		KSC	10 ^c–e^	13 ^b–e^	15 ^a–e^	13 ^b–e^
		KSP	10 ^c–e^	10 ^c–e^	13 ^b–e^	12 ^c–e^
		NqSC	8 ^e^	9 ^de^	11 ^c–e^	15 ^a–e^
	**Fruits**		**7 DAA**	**14 DAA**	**21 DAA**	
		BG-24	9.21 ^e–i^	3.60 ^pq^	7.41 ^i–n^	
		BG-27	10.20 ^c–g^	10.78 ^c–f^	11.62 ^cd^	
		BG-31	5.40 ^m–p^	7.59 ^h–m^	6.61 ^j–o^	
		BG-100/GC	11.23 ^c–e^	5.60 ^l–p^	0.70 ^rs^	
		DSI	6.99 ^i–n^	8.79 ^f–j^	12.17 ^c^	
		ESC	12.29 ^c^	6.99 ^i–n^	3.60 ^pq^	
		KRI	10.19 ^c–g^	5.20 ^nop^	7.20 ^i–n^	
		KSC	11.20 ^c–e^	22.62 ^a^	4.40 ^o–q^	
		KSP	9.21 ^e–i^	7.80 ^h–l^	8.00 ^g–k^	
		NqRC	0.60 ^rs^	3.81 ^pq^	6.59 ^j–o^	
		NqSC	0.50 ^s^	6.81 ^j–n^	8.00 ^g–k^	
		NRC	9.82 ^d–h^	6.80 ^j–n^	6.21 ^k–o^	
		NSRC	2.60 ^qrs^	7.01 ^i–n^	6.59 ^j–o^	
		NSRP	8.40 ^g–k^	8.00 ^g–k^	17.43 ^b^	
		RSP	0.50 ^s^	0.90 ^rs^	2.80 ^qr^	
**Al (mg/kg)**	**Fruits**		**7 DAA**	**14 DAA**	**21 DAA**	
		BG-24	38.37 ^a^	38.84 ^a^	39.17 ^a^	
		BG-27	40.24 ^a^	40.25 ^a^	39.39 ^a^	
		BG-31	36.78 ^a^	38.50 ^a^	38.89 ^a^	
		BG-100/GC	37.53 ^a^	38.73 ^a^	35.64 ^a^	
		DSI	33.70 ^a^	38.24 ^a^	39.94 ^a^	
		ESC	35.99 ^a^	36.90 ^a^	41.10 ^a^	
		KRI	37.10 ^a^	37.20 ^a^	38.90 ^a^	
		KSC	37.99 ^a^	35.35 ^a^	42.66 ^a^	
		KSP	36.90 ^a^	37.70 ^a^	38.00 ^a^	
		NqRC	36.09 ^a^	39.39 ^a^	37.60 ^a^	
		NqSC	37.13 ^a^	38.86 ^a^	39.90 ^a^	
		NRC	37.20 ^a^	37.30 ^a^	38.30 ^a^	
		NSRC	35.80 ^a^	37.60 ^a^	39.50 ^a^	
		NSRP	37.80 ^a^	42.00 ^a^	41.50 ^a^	
		RSP	35.10 ^a^	34.80 ^a^	41.40 ^a^	

Means followed by different superscript letters, both lowercase and uppercase after “z”, show significant differences in traits within and across growth stages (columns) for each landrace. The letters indicate the level of significance among landraces during various growth phases. This is based on Tukey’s honest significant difference test, with a significance level of *p* < 0.05. Landraces are explained in Table 2. Mic—micronutrients, DAS—days after sowing, DAA—days after anthesis, Na—sodium, Mn—manganese, Fe—iron, Zn—zinc, Cu—copper, and Al—aluminum in mg/kg.

**Table 4 plants-13-01475-t004:** Proximates of *Lagenaria siceraria* shoots and fruits harvested at different stages of growth.

Proximate	Plant Part	Landraces		Harvesting Period		
**MC (g/100 g)**	**Shoots**		**42 DAS**	**49 DAS**	**56 DAS**	**63 DAS**
		BG-19	9.02 ^a–h^	10.31 ^a–f^	8.24 ^a–i^	8.52 ^a–i^
		BG-24	8.89 ^a–h^	10.2 ^a–g^	10.23 ^a–g^	9.68 ^a–h^
		BG-26	10.79 ^a–e^	10.98 ^a–d^	6.4 ^d–i^	4.08 ^i^
		BG-27	5.25 ^hi^	6.14 ^e–i^	5.63 ^f–i^	5.55 ^g–i^
		BG-80	7.48 ^b–i^	9.40 ^a–h^	10.26 ^a–g^	9.39 ^a–h^
		BG-81	9.09 ^a–h^	8.09 ^a–i^	8.86 ^a–h^	8.26 ^a–i^
		BG-100/GC	7.41 ^b–i^	7.02 ^c–i^	7.44 ^b–i^	6.39 ^d–i^
		DSI	10.81 ^a–e^	12.68 ^a^	9.25 ^a–h^	8.04 ^a–i^
		ESC	10.69 ^a–e^	8.63 ^a–i^	10.55 ^a–g^	9.80 ^a–h^
		KRI	8.56 ^a–i^	11.47 ^a–c^	8.89 ^a–h^	9.64 ^a–h^
		KSC	12.02 ^ab^	10.66 ^a–e^	8.87 ^a–h^	7.45 ^a–i^
		KSP	9.51 ^a–h^	12.05 ^ab^	9.65 ^a–h^	7.82 ^b–i^
		NqSC	12.11 ^ab^	11.09 ^a–d^	10.69 ^a–e^	10.63 ^a–e^
	**Fruits**		**7 DAA**	**14 DAA**	**21 DAA**	
		BG-24	12.18 ^d–j^	9.00 ^h–p^	10.51 ^f–n^	
		BG-27	6.68 ^l–q^	5.96 ^n–q^	8.33 ^j–q^	
		BG-31	12.24 ^c–j^	8.61 ^i–p^	8.32 ^j–q^	
		BG-100/GC	7.90 ^j–q^	13.83 ^b–g^	18.87 ^a^	
		DSI	7.46 ^k–q^	9.62 ^g–o^	5.88 ^n–q^	
		ESC	12.37 ^b–j^	11.33 ^e–l^	3.76 ^q^	
		KRI	14.29 ^a–f^	10.24 ^f–o^	4.51 ^pq^	
		KSC	16.77 ^a–d^	12.28 ^c–j^	8.17 ^j–q^	
		KSP	17.01 ^ab^	13.50 ^b–h^	9.69 ^f–o^	
		NqRC	15.58 ^a–e^	13.86 ^b–g^	6.57 ^m–q^	
		NqSC	15.43 ^a–e^	13.06 ^b–i^	4.71 ^pq^	
		NRC	11.15 ^e–m^	11.02 ^e–m^	10.66 ^f–m^	
		NSRC	16.87 ^a–c^	14.23 ^a–g^	8.29 ^j–q^	
		NSRP	11.54 ^e–k^	4.89 ^pq^	9.05 ^h–p^	
		RSP	11.40 ^e–k^	9.17 ^h–p^	5.64 ^o–q^	
**ADF (g/100 g)**	**Shoots**		**42 DAS**	**49 DAS**	**56 DAS**	**63 DAS**
		BG-19	27.94 ^g–q^	23.30 ^l–q^	24.09 ^k–q^	36.42 ^b–i^
		BG-24	20.57 ^n–q^	23.40 ^l–q^	22.89 ^m–q^	37.00 ^b–h^
		BG-26	18.40 ^q^	19.61 ^o–q^	32.32 ^e–n^	37.90 ^b–g^
		BG-27	18.90 ^pq^	18.59 ^q^	26.87 ^g–q^	30.84 ^f–o^
		BG-80	19.04 ^o–q^	26.01 ^h–q^	26.68 ^g–q^	41.31 ^a–f^
		BG-81	27.80 ^g–q^	24.68 ^i–q^	35.52 ^b–k^	35.99 ^b–j^
		BG-100/GC	28.12 ^g–q^	23.84 ^k–q^	32.26 ^e–n^	43.03 ^a–e^
		DSI	24.55 ^j–q^	23.52 ^l–q^	34.95 ^c–l^	44.17 ^a–d^
		ESC	20.77 ^m–q^	22.02 ^m–q^	29.49 ^f–q^	37.06 ^b–h^
		KRI	31.54 ^e–n^	29.63 ^f–q^	36.07 ^b–j^	32.41 ^d–m^
		KSC	25.05 ^i–q^	21.36 ^m–q^	30.62 ^f–p^	46.85 ^ab^
		KSP	21.52 ^m–q^	21.09 ^m–q^	31.89 ^e–n^	50.63 ^a^
		NqSC	26.19 ^g–q^	26.19 ^g–q^	29.04 ^g–q^	45.24 ^a–c^
	**Fruits**		**7 DAA**	**14 DAA**	**21 DAA**	
		BG-24	11.43 ^j^	15.63 ^g–j^	21.18 ^e–j^	
		BG-27	14.49 ^ij^	26.79 ^d–h^	18.91 ^e–j^	
		BG-31	12.29 ^j^	18.72 ^e–j^	21.33 ^e–j^	
		BG-100/GC	13.24 ^ij^	12.60 ^ij^	18.73 ^e–j^	
		DSI	27.27 ^d–f^	12.22 ^j^	19.32 ^e–j^	
		ESC	12.50 ^ij^	15.21 ^h–j^	60.25 ^a^	
		KRI	24.05 ^d–i^	14.23 ^ij^	27.07 ^d–g^	
		KSC	14.03 ^ij^	35.29 ^cd^	57.04 ^a^	
		KSP	18.16 ^f–j^	16.18 ^f–j^	17.60 ^f–j^	
		NqRC	18.84 ^e–j^	17.32 ^f–j^	19.17 ^e–j^	
		NqSC	20.61 ^e–j^	14.16 ^ij^	27.69 ^d–f^	
		NRC	14.73 ^ij^	13.98 ^ij^	21.29 ^e–j^	
		NSRC	17.28 ^f–j^	18.82 ^e–j^	35.29 ^cd^	
		NSRP	13.86 ^ij^	42.68 ^bc^	50.46 ^ab^	
		RSP	12.62 ^ij^	30.12 ^de^	57.78 ^a^	
**NDF (g/100 g)**	**Shoots**		**42 DAS**	**49 DAS**	**56 DAS**	**63 DAS**
		BG-19	34.69 ^f–q^	29.54 ^m–q^	31.80 ^i–q^	42.73 ^c–j^
		BG-24	30.47 ^k–q^	33.28 ^g–q^	30.71 ^k–q^	49.16 ^a–e^
		BG-26	25.37 ^q^	27.48 ^o–q^	38.38 ^e–p^	44.95 ^b–g^
		BG-27	25.18 ^q^	26.86 ^pq^	35.53 ^f–q^	39.73 ^c–n^
		BG-80	25.15 ^q^	32.94 ^h–q^	35.23 ^f–q^	50.99 ^a–d^
		BG-81	34.80 ^f–q^	32.45 ^h–q^	41.85 ^c–k^	43.80 ^c–h^
		BG-100/GC	38.34 ^e–p^	33.78 ^f–q^	42.81 ^c–j^	50.06 ^a–e^
		DSI	31.37 ^j–q^	29.87 ^l–q^	44.14 ^c–h^	56.65 ^ab^
		ESC	30.76 ^k–q^	29.46 ^m–q^	34.61 ^f–q^	45.41 ^b–f^
		KRI	38.60 ^e–p^	39.74 ^c–n^	43.30 ^c–i^	40.55 ^c–m^
		KSC	33.98 ^f–q^	31.76 ^i–q^	39.19 ^d–o^	56.06 ^ab^
		KSP	28.28 ^n–q^	32.46 ^h–q^	41.65 ^c–l^	59.06 ^a^
		NqSC	32.54 ^h–q^	32.71 ^h–q^	36.02 ^f–q^	51.26 ^a–c^
	**Fruits**		**7 DAA**	**14 DAA**	**21 DAA**	
		BG-100/GC	22.54 ^i–m^	19.29 ^k–m^	20.88 ^j–m^	
		BG-24	20.98 ^j–m^	24.61 ^h–m^	40.05 ^c–f^	
		BG-27	23.63 ^h–m^	42.12 ^cd^	27.42 ^g–m^	
		BG-31	19.29 ^k–m^	25.36 ^h–m^	32.45 ^d–j^	
		DSI	33.09 ^c–i^	20.95 ^j–m^	34.31 ^c–h^	
		ESC	25.63 ^g–m^	25.50 ^g–m^	73.31 ^a^	
		KRI	29.04 ^f–l^	17.71 ^lm^	41.24 ^c–e^	
		KSC	21.95 ^i–m^	39.63 ^c–f^	68.74 ^a^	
		KSP	23.37 ^h–m^	20.05 ^k–m^	23.20 ^h–m^	
		NqRC	23.02 ^h–m^	21.30 ^j–m^	25.96 ^g–m^	
		NqSC	22.10 ^i–m^	21.24 ^j–m^	37.07 ^c–g^	
		NRC	17.08 ^m^	16.31 ^m^	30.13 ^e–k^	
		NSRC	18.97 ^k–m^	23.2 ^h–m^	43.58 ^cd^	
		NSRP	23.69 ^h–m^	56.53 ^b^	67.49 ^ab^	
		RSP	21.63 ^i–m^	44.25 ^c^	71.51 ^a^	
**ADL (g/100 g)**	**Shoots**		**42 DAS**	**49 DAS**	**56 DAS**	**63 DAS**
		BG-19	8.61 ^c–f^	4.40 ^i–s^	6.76 ^e–i^	14.68 ^b^
		BG-24	2.92 ^r–t^	4.00 ^l–t^	3.30 ^n–t^	6.75 ^e–i^
		BG-26	4.02 ^l–t^	3.09 ^p–t^	4.27 ^j–t^	8.44 ^c–f^
		BG-27	3.26 ^n–t^	4.14 ^k–t^	4.48 ^h–s^	6.40 ^f–k^
		BG-80	1.99 ^t^	3.93 ^l–t^	5.43 ^g–p^	9.11 ^c–e^
		BG-81	2.85 ^st^	3.23 ^o–t^	3.28 ^n–t^	7.55 ^d–j^
		BG-100/GC	2.46 ^st^	2.89 ^st^	3.06 ^q–t^	6.82 ^e–h^
		DSI	7.06 ^d–g^	3.90 ^l–t^	5.62 ^g–n^	6.61 ^f–j^
		ESC	3.40 ^m–t^	3.69 ^l–t^	5.72 ^g–m^	10.31 ^c^
		KRI	12.82 ^b^	6.01 ^g–l^	23.34 ^a^	14.06 ^b^
		KSC	5.61 ^g–n^	2.66 ^st^	5.49 ^g–o^	5.31 ^g–q^
		KSP	5.28 ^g–r^	4.05 ^k–t^	4.30 ^j–t^	9.29 ^cd^
		NqSC	5.46 ^g–o^	5.40 ^g–q^	9.19 ^cd^	13.99 ^b^
**Ash (g/100 g)**	**Fruits**		**7 DAA**	**14 DAA**	**21 DAA**	
		BG-24	5.46 ^j–r^	4.23 ^o–s^	6.27 ^h–p^	
		BG-27	4.34 ^n–s^	7.65 ^e–j^	6.58 ^f–n^	
		BG-31	4.70 ^l–s^	2.53 ^s^	3.84 ^qrs^	
		BG-100/GC	10.33 ^d^	3.78 ^qrs^	8.82 ^d–g^	
		DSI	35.60 ^a^	5.98 ^h–q^	5.40 ^j–r^	
		ESC	4.17 ^p–s^	7.08 ^f–k^	8.73 ^d–g^	
		KRI	6.63 ^f–n^	4.32 ^n–s^	4.82 ^k–s^	
		KSC	6.89 ^f–m^	9.93 ^de^	7.46 ^f–j^	
		KSP	7.94 ^e–i^	6.50 ^g–p^	4.22 ^o–s^	
		NqRC	19.91 ^c^	4.59 ^m–s^	5.32 ^j–r^	
		NqSC	21.39 ^c^	5.47 ^j–r^	7.04 ^f–k^	
		NRC	5.64 ^i–r^	4.70 ^l–s^	4.84 ^k–s^	
		NSRC	6.55 ^g–o^	6.62 ^f–n^	5.02 ^k–r^	
		NSRP	6.10 ^h–q^	7.98 ^e–h^	8.88 ^d–f^	
		RSP	3.40 ^rs^	26.11 ^b^	7.02 ^f–l^	
**Protein (g/100 g)**	**Shoots**		**42 DAS**	**49 DAS**	**56 DAS**	**63 DAS**
		BG-19	22.20 ^ab^	20.49 ^ab^	24.23 ^ab^	21.87 ^ab^
		BG-24	18.52 ^ab^	19.67 ^ab^	20.76 ^ab^	26.65 ^ab^
		BG-26	24.83 ^ab^	24.69 ^ab^	22.77 ^ab^	20.36 ^ab^
		BG-27	20.40 ^ab^	17.75 ^b^	22.20 ^ab^	19.31 ^ab^
		BG-80	21.29 ^ab^	20.46 ^ab^	24.45 ^ab^	23.21 ^ab^
		BG-81	19.12 ^ab^	21.13 ^ab^	20.65 ^ab^	23.54 ^ab^
		BG-100/GC	18.14 ^ab^	18.86 ^ab^	20.74 ^ab^	22.54 ^ab^
		DSI	25.50 ^ab^	24.42 ^ab^	21.45 ^ab^	24.09 ^ab^
		ESC	23.05 ^ab^	20.65 ^ab^	22.44 ^ab^	21.81 ^ab^
		KRI	29.70 ^a^	21.94 ^ab^	27.23 ^ab^	26.29 ^ab^
		KSC	22.80 ^ab^	22.34 ^ab^	19.12 ^ab^	23.78 ^ab^
		KSP	22.55 ^ab^	25.92 ^ab^	22.82 ^ab^	18.69 ^ab^
		NqSC	26.34 ^ab^	26.78 ^ab^	20.48 ^ab^	24.74 ^ab^
	**Fruits**		**7 DAA**	**14 DAA**	**21 DAA**	
		BG-24	10.60 ^b–j^	7.85 ^d–l^	7.66 ^e–l^	
		BG-27	9.67 ^b–k^	5.74 ^kl^	8.73 ^c–k^	
		BG-31	6.55 ^i–l^	8.78 ^c–k^	6.45 ^i–l^	
		BG-100/GC	12.15 ^b–e^	21.15 ^a^	5.03 ^kl^	
		DSI	7.25 ^f–l^	6.11 ^i–l^	7.89 ^d–l^	
		ESC	9.30 ^b–k^	11.72 ^b–f^	5.51 ^kl^	
		KRI	12.69 ^bc^	6.07 ^jkl^	5.04 ^kl^	
		KSC	12.17 ^b–e^	3.71 ^l^	9.53 ^b–k^	
		KSP	10.75 ^b–i^	11.63 ^b–g^	6.09 ^i–l^	
		NqRC	7.32 ^f–l^	11.43 ^b–h^	6.44 ^i–l^	
		NqSC	7.21 ^f–l^	8.39 ^c–k^	6.81 ^h–l^	
		NRC	8.19 ^c–l^	7.00 ^g–l^	5.38 ^kl^	
		NSRC	6.63 ^i–l^	12.38 ^b–d^	7.25 ^f–l^	
		NSRP	13.56 ^b^	8.00 ^d–l^	7.45 ^f–l^	
		RSP	7.56 ^e–l^	6.69 ^i–l^	5.58 ^kl^	
**Carbohydrates (g/100 g)**	**Shoots**		**42 DAS**	**49 DAS**	**56 DAS**	**63 DAS**
		BG-19	9.14 ^o–s^	10.85 ^h–l^	9.57 ^m–q^	7.01 ^v–x^
		BG-24	12.09 ^e–g^	8.45 ^q–u^	10.63 ^j–m^	5.96 ^x–z^
		BG-26	14.90 ^a–c^	14.67 ^bc^	9.89 ^k–p^	6.44 ^w–y^
		BG-27	15.79 ^ab^	13.91 ^cd^	11.07 ^g–k^	9.80 ^l–p^
		BG-80	9.34 ^n–r^	10.31 ^j–o^	7.54 ^u–w^	3.86 ^C^
		BG-81	5.65 ^yzA^	11.98 ^e–h^	7.79 ^t–v^	6.40 ^w–y^
		BG-100/GC	13.08 ^de^	8.03 ^s–v^	5.80 ^yz^	4.49 ^ABC^
		DSI	11.30 ^g–j^	9.85 ^l–p^	8.94 ^p–t^	4.28 ^BC^
		ESC	11.84 ^f–i^	15.87 ^a^	11.82 ^f–i^	8.99 ^p–s^
		KRI	1.61 ^D^	8.05 ^s–v^	1.35 ^D^	3.67 ^C^
		KSC	12.23 ^e–g^	12.60 ^ef^	10.44 ^j–n^	5.68 ^yz^
		KSP	12.72 ^ef^	11.95 ^e–h^	10.80 ^h–l^	5.07 ^zAB^
		NqSC	8.11 ^s–v^	8.32 ^r–u^	10.73 ^i–m^	7.43 ^u–w^
**Fat (g/100 g)**	**Fruits**		**7 DAA**	**14 DAA**	**21 DAA**	
		BG-24	3.42 ^g^	0.50 ^r–t^	0.65 ^q–t^	
		BG-27	0.54 ^q–t^	0.57 ^q–t^	0.35 ^st^	
		BG-31	0.75 ^o–s^	3.21 ^gh^	3.00 ^g–i^	
		BG-100/GC	4.64 ^d^	4.15 ^ef^	3.92 ^f^	
		DSI	0.91 ^m–r^	0.83 ^n–r^	3.90 ^f^	
		ESC	3.00 ^g–i^	0.99 ^m–q^	0.24 ^t^	
		KRI	4.00 ^f^	4.03 ^f^	1.70 ^jk^	
		KSC	4.58 ^de^	5.15 ^c^	0.81 ^n–s^	
		KSP	1.27 ^k–n^	2.88 ^hi^	2.59 ^i^	
		NqRC	1.16 ^l–p^	0.73 ^p–s^	0.76 ^o–s^	
		NqSC	0.48 ^r–t^	0.84 ^n–r^	1.21 ^l–o^	
		NRC	3.32 ^gh^	1.6 ^j–l^	5.73 ^b^	
		NSRC	0.56 ^q–t^	0.87 ^m–r^	3.37 ^g^	
		NSRP	1.85 ^j^	6.92 ^a^	3.04 ^g–i^	
		RSP	5.18 ^c^	1.31 ^k–m^	0.53 ^q–t^	

Means followed by different superscript letters, both lowercase and uppercase after “z”, show significant differences in traits within and across growth stages (columns) for each landrace. The letters indicate the level of significance among landraces during various growth phases. This is based on Tukey’s honest significant difference test, with a significance level of *p* < 0.05. Landraces are explained in Table 2. DAS—days after sowing, DAA—days after anthesis, MC—moisture content, ADF—acid detergent fiber, NDF—neutral detergent fiber, and ADL—acid detergent lignin in g/100 g.

**Table 5 plants-13-01475-t005:** Correlation among the nutritional and mineral composition of the *L. siceraria* shoots at different growth stages.

Var	Ca	Mg	K	P	N	Na	Mn	Fe	Zn	Cu	MC	ADF	NDF	K/Ca+Mg	ADL	Carbs
Mg	**0.709**															
K	0.552	**0.879**														
P	0.531	**0.759**	**0.690**													
N	0.484	**0.746**	**0.858**	**0.615**												
Na	0.170	−0.246	−0.062	−0.042	−0.098											
Mn	0.155	−0.290	−0.238	0.191	−0.213	**0.644**										
Fe	−0.483	**−0.659**	**−0.660**	−0.540	−0.584	0.142	0.287									
Zn	−0.328	−0.545	−0.462	0.035	−0.304	0.401	**0.767**	0.406								
Cu	−0.166	−0.032	−0.021	0.339	−0.107	−0.077	0.272	0.077	0.324							
MC	0.340	0.329	0.514	0.093	**0.662**	0.264	−0.115	−0.317	−0.141	−0.104						
ADF	0.189	0.301	0.381	−0.033	0.496	−0.278	−0.301	0.172	−0.364	0.019	0.490					
NDF	0.060	0.222	0.223	−0.143	0.339	−0.359	−0.427	0.231	−0.428	0.140	0.421	**0.933**				
K/Ca+Mg	**−0.735**	−0.358	0.042	−0.179	0.023	0.019	−0.095	0.168	0.230	0.241	0.049	0.087	0.048			
ADL	**0.613**	**0.889**	**0.921**	**0.786**	**0.817**	−0.170	−0.155	**−0.687**	−0.347	0.091	0.375	0.359	0.204	−0.080		
Carbs	−0.201	−0.461	−0.480	−0.320	−0.556	**0.617**	0.293	0.324	0.259	−0.324	−0.298	**−0.661**	**−0.614**	−0.085	**−0.631**	
Protein	0.484	**0.746**	**0.858**	**0.614**	**1.000**	−0.097	−0.213	−0.584	−0.304	−0.107	**0.663**	0.497	0.340	0.024	**0.817**	−0.556

Values ≥ 0.6 are deemed to be significantly correlated and are in bold. Var, variables—Ca, calcium (g/100 g); Mg, magnesium (g/100 g); K, potassium (g/100 g); P, phosphorus (g/100 g); N, nitrogen (g/100 g); Na, sodium (mg/kg); Mn, manganese (mg/kg); Fe, iron (mg/kg); Zn, zinc (mg/kg); Cu, copper (mg/kg); MC, moisture content (g/100 g); ADF, acid detergent fiber (g/100 g); NDF, neutral detergent fiber (g/100 g); K/Ca+Mg, potassium/calcium+magnesium ratio (g/100 g); ADL, acid detergent lignin (g/100 g); Carbs, carbohydrates (g/100 g); and protein (g/100 g).

**Table 6 plants-13-01475-t006:** Correlation among the nutritional and mineral composition of the *L. siceraria* fruits at different growth stages.

Var	Ca	Mg	K	P	N	C	C/N	Na	Mn	Fe	Zn	Cu	Al	Ash	MC	ADF	NDF	Protein
Mg	−0.224																	
K	0.535	−0.040																
P	−0.265	0.533	−0.368															
N	0.321	0.228	0.149	0.544														
C	0.037	0.566	−0.183	**0.600**	0.333													
C/N	−0.270	−0.293	−0.242	−0.339	**−0.877**	−0.303												
Na	0.050	0.252	0.330	0.265	−0.180	0.315	0.190											
Mn	0.524	−0.479	0.057	0.069	0.481	−0.330	−0.272	−0.205										
Fe	0.382	0.069	0.107	0.552	**0.810**	0.148	−0.546	−0.043	**0.727**									
Zn	0.205	0.201	0.271	0.576	**0.726**	0.077	−0.582	0.234	0.536	**0.855**								
Cu	0.053	0.481	−0.071	**0.864**	**0.673**	**0.602**	−0.521	0.313	0.085	0.555	**0.610**							
Al	0.440	0.060	0.136	0.518	**0.781**	0.132	−0.517	−0.009	**0.747**	**0.996**	**0.843**	0.531						
Ash	0.439	−0.142	**0.837**	−0.531	0.112	−0.233	−0.285	0.125	−0.039	−0.035	0.075	−0.192	−0.008					
MC	−0.112	−0.459	−0.443	−0.072	0.269	−0.397	−0.237	**−0.749**	0.549	0.230	0.042	−0.212	0.202	−0.240				
ADF	**0.894**	−0.008	0.428	0.003	0.406	0.325	−0.234	0.144	0.422	0.472	0.267	0.242	0.514	0.217	−0.254			
NDF	**0.816**	0.079	0.504	0.051	0.304	0.426	−0.163	0.333	0.247	0.374	0.247	0.265	0.414	0.214	−0.454	**0.955**		
Protein	0.323	0.227	0.152	0.545	**1.000**	0.330	**−0.874**	−0.175	0.485	**0.813**	**0.730**	**0.673**	**0.786**	0.112	0.266	0.408	0.306	
Fat	0.128	−0.362	−0.080	0.142	0.224	−0.083	0.034	−0.326	0.307	0.312	0.111	0.265	0.284	−0.191	0.238	0.225	0.156	0.221

Values ≥ 0.6 are deemed to be significantly correlated and are in bold. Var, variables—Ca, calcium (g/100 g); Mg, magnesium (g/100 g); K, potassium (g/100 g); P, phosphorus (g/100 g); N, nitrogen (g/100 g); C, carbon (g/100 g); C/N, carbon/nitrogen ratio (g/100 g); Na, sodium (mg/kg); Mn, manganese (mg/kg); Fe, iron (mg/kg); Zn, zinc (mg/kg); Cu, copper (mg/kg); Al, aluminum (mg/kg); MC, moisture content (g/100 g); ADF, acid detergent fiber (g/100 g); NDF, neutral detergent fiber (g/100 g); protein (g/100 g); and fat (g/100 g).

**Table 7 plants-13-01475-t007:** Loadings of the shoot nutrient traits for the first five principal components.

Variables	PC1	PC2	PC3	PC4	PC5
Ca	**0.632**	−0.468	−0.249	0.280	−0.457
Mg	**0.910**	−0.203	−0.191	−0.126	−0.117
K	**0.921**	−0.158	0.072	−0.007	0.202
P	**0.646**	−0.555	0.227	−0.372	−0.119
N	**0.912**	−0.057	0.184	0.148	0.202
Na	−0.264	−0.582	0.268	**0.621**	0.128
Mn	−0.348	**−0.641**	0.514	0.174	−0.317
Fe	**−0.670**	0.371	0.278	0.250	−0.326
Zn	−0.533	−0.380	**0.643**	−0.021	−0.039
Cu	−0.025	0.001	**0.642**	−0.517	−0.304
Moisture	0.564	0.067	0.270	**0.608**	0.207
ADF	0.536	**0.651**	0.266	0.327	−0.270
NDF	0.423	**0.770**	0.190	0.240	−0.322
K/Ca+Mg	−0.155	0.342	0.562	−0.171	**0.673**
ADL	**0.922**	−0.205	0.106	−0.174	0.007
Carbohydrates	**−0.673**	−0.424	−0.284	0.304	0.213
Protein	**0.912**	−0.056	0.184	0.149	0.203
Eigenvalue	7.193	2.991	2.049	1.710	1.395
Variability (%)	42.312	17.594	12.050	10.058	8.204
Cumulative (%)	42.312	59.906	71.956	82.014	90.218

PC1–5: principal components 1–5. Values ≥ 0.6 are deemed to be significant and are in bold. Variables—Ca, calcium (g/100 g); Mg, magnesium (g/100 g); K, potassium (g/100 g); P, phosphorus (g/100 g); N, nitrogen (g/100 g); Na, sodium (mg/kg); Mn, manganese (mg/kg); Fe, iron (mg/kg); Zn, zinc (mg/kg); Cu, copper (mg/kg); MC, moisture content (g/100 g); ADF, acid detergent fiber (g/100 g); NDF, neutral detergent fiber (g/100 g), K/Ca+Mg, potassium/calcium+magnesium ratio (g/100 g); ADL, acid detergent lignin (g/100 g); Carbs, carbohydrates (g/100 g); and protein (g/100 g).

**Table 8 plants-13-01475-t008:** Loadings of the fruit nutrient traits for the first five principal components.

Variables	PC1	PC2	PC3	PC4	PC5
Ca	0.505	**0.759**	−0.088	−0.208	−0.193
Mg	0.214	−0.514	**−0.611**	0.247	−0.150
K	0.237	**0.737**	−0.359	0.379	0.164
P	0.582	**−0.751**	−0.181	−0.158	0.163
N	**0.920**	−0.127	0.148	0.245	−0.190
C	0.368	−0.413	**−0.603**	−0.206	−0.420
C/N ratio	**−0.731**	0.083	−0.062	−0.545	0.311
Na	0.059	0.029	**−0.759**	−0.102	0.555
Mn	0.585	0.310	**0.604**	−0.206	0.261
Fe	**0.915**	−0.030	0.233	−0.069	0.214
Zn	**0.827**	−0.094	0.036	0.190	0.466
Cu	**0.721**	−0.476	−0.298	−0.073	0.035
Al	**0.910**	0.025	0.209	−0.091	0.232
Ash	0.076	**0.700**	−0.200	0.595	−0.006
MC	0.068	−0.130	**0.936**	0.094	−0.160
ADF	**0.617**	0.552	−0.255	−0.400	−0.244
NDF	0.548	0.515	−0.460	−0.405	−0.175
Protein	**0.922**	−0.124	0.147	0.243	−0.183
Fat	0.268	0.033	0.391	−0.495	−0.090
Eigenvalue	7.043	3.568	3.428	1.770	1.275
Variability (%)	37.069	18.781	18.042	9.317	6.709
Cumulative %	37.069	55.850	73.893	83.209	89.918

PC1–5: principal components 1–5. Values ≥ 0.6 are deemed to be significant and are in bold. Variables—Ca, calcium (g/100 g); Mg, magnesium (g/100 g); K, potassium (g/100 g); P, phosphorus (g/100 g); N, nitrogen (g/100 g); C, carbon (g/100 g); C/N, carbon/nitrogen ratio (g/100 g); Na, sodium (mg/kg); Mn, manganese (mg/kg); Fe, iron (mg/kg); Zn, zinc (mg/kg); Cu, copper (mg/kg); Al, aluminum (mg/kg); MC, moisture content (g/100 g); ADF, acid detergent fiber (g/100 g); NDF, neutral detergent fiber (g/100 g); protein (g/100 g); and fat (g/100 g).

## Data Availability

The research data can be requested from the authors.

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
