# Peer review of "Nutritional Variation on Sequentially Harvested Shoots and Fruits of Lagenaria siceraria Landraces"

_plants, 2024, doi:10.3390/plants13111475_

Round 1

Reviewer 1 Report

Comments and Suggestions for Authors

The article submitted for review is very interesting, innovative and well-written. The results are supported by well-selected recent literature. The plant included in the research, although known, is presented in a new light for practical use. The article is of a high substantive level, but it is surprising that so many results are included in one article. For me, the material is too extensive for an article, too many results are difficult to read and interpret for a scientific article and not a monograph. I suggest dividing the article into two parts, either fruits and shoots separately, or presenting macro and micronutrients separately and making two articles. I also don't understand why the tables do not provide units, which is illegible. Although the article is good, I suggest dividing it into two parts, so I reject it in this form

Author Response

Reviewer 1

General reviewer’s comments

Item

Comment

Response

1

The article is of a high substantive level, but it is surprising that so many results are included in one article. For me, the material is too extensive for an article, too many results are difficult to read and interpret for a scientific article and not a monograph. I suggest dividing the article into two parts, either fruits and shoots separately, or presenting macro and micronutrients separately and making two articles.

The article has not been separated into two manuscripts because the shoot and fruit material were harvested from the same plants and cannot be treated as separate entities. By not separating the data, we were able to observe the effect of redirected assimilates by the nutrients present from different plant parts (shoots to fruits and vice versa) at varying growth stages, which was the aim of the study. To date, no study has investigated L. siceraria in this manner, hence the comprehensive results yielded from our research. As a single manuscript, this approach perfectly encapsulates the ideal timing for harvesting the most nutritious vegetative modules and fruits of L. siceraria.

2

I also don't understand why the tables do not provide units, which is illegible.

The units were provided in the table footnotes (g/100g and mg/kg); however, they have since been incorporated into the tables and footnotes for each nutrient trait.  

Page 4, line 93 – 124

I also don't understand why the tables do not provide units, which is illegible.

Corrected as - Mac-macronutrients, DAS- days after sowing, and DAA- days after anthesis. Ca-calcium, Mg-magnesium, K-potassium, P-phosphorus, N-nitrogen, C-carbon, C/N ratio- carbon/nitrogen ratio, and K/Ca+Mg -potassium/calcium+magnesium ratio in g/100g

Page 13, line 270 – 297

I also don't understand why the tables do not provide units, which is illegible.

Corrected as - Mic-micronutrients, DAS- days after sowing, and DAA- days after anthesis. Na-sodium, Mn-manganese, Fe-iron, Zn-zinc, Cu-copper, and Al-aluminium in mg/kg.

Page 22, line 470 – 498

I also don't understand why the tables do not provide units, which is illegible.

Corrected as - MC- Moisture content, ADF- acid detergent fibre, NDF- neutral detergent fibre, and ADL-Acid Detergent lignin in g/100g.

Page 31, line 669 – 675 and 677 – 685

I also don't understand why the tables do not provide units, which is illegible.

Table 5a Corrected as – Values ≥ 0.6 are deemed to be significantly correlated and are in bold. Var, Variables -Ca, calcium (g/100g); Mg, magnesium (g/100g); K, potassium (g/100g); P, phosphorus (g/100g); N, nitrogen (g/100g); Na, sodium (mg/kg); Mn, manganese (mg/kg); Fe, iron (mg/kg); Zn, zinc (mg/kg); Cu, copper (mg/kg); MC, moisture content (g/100g); ADF, acid detergent fibre (g/100g); NDF. Neutral detergent fibre (g/100g), K/Ca+Mg -potassium/calcium+magnesium ratio(g/100g); ADL, acid detergent lignin (g/100g); Carbs, carbohydrates (g/100g); and protein (g/100g).

Table 5b Corrected as - Values ≥ 0.6 are deemed to be significantly correlated and are in bold. Var, Variables –Ca, calcium (g/100g); Mg, magnesium (g/100g); K, potassium (g/100g); P, phosphorus (g/100g); N, nitrogen  (g/100g); C, carbon (g/100g); C/N, carbon/nitrogen ratio (g/100g); Na, sodium (mg/kg); Mn, manganese (mg/kg); Fe, iron (mg/kg); Zn, zinc (mg/kg); Cu, copper (mg/kg); Al, aluminium (mg/kg); MC, moisture content (g/100g); ADF, acid detergent fibre (g/100g); and NDF, Neutral detergent fibre (g/100g); Protein (g/100g); and Fat (g/100g).

Page 32, line 700 – 706

I also don't understand why the tables do not provide units, which is illegible

Table 6a corrected as - PC1–5: Principal components 1–5. Values ≥ 0.6 are deemed to be significant and are in bold. Variables –Ca, calcium (g/100g); Mg, magnesium (g/100g); K, potassium (g/100g); P, phosphorus (g/100g); N, nitrogen (g/100g); Na, sodium (mg/kg); Mn, manganese (mg/kg); Fe, iron (mg/kg); Zn, zinc (mg/kg); Cu, copper (mg/kg); MC, moisture content (g/100g); ADF, acid detergent fibre (g/100g); NDF. Neutral detergent fibre (g/100g), K/Ca+Mg -potassium/calcium+magnesium ratio(g/100g); ADL, acid detergent lignin (g/100g); Carbs, carbohydrates (g/100g); and protein (g/100g).

Page 33, line 727 – 733

I also don't understand why the tables do not provide units, which is illegible

Table 6b corrected as - PC1–5: Principal components 1–5. Values ≥ 0.6 are deemed to be significant and are in bold. Variables –Ca, calcium (g/100g); Mg, magnesium (g/100g); K, potassium (g/100g); P, phosphorus (g/100g); N, nitrogen  (g/100g); C, carbon (g/100g); C/N, carbon/nitrogen ratio (g/100g); Na, sodium (mg/kg); Mn, manganese (mg/kg); Fe, iron (mg/kg); Zn, zinc (mg/kg); Cu, copper (mg/kg); Al, aluminium (mg/kg); MC, moisture content (g/100g); ADF, acid detergent fibre (g/100g); and NDF, Neutral detergent fibre (g/100g); Protein (g/100g); and Fat (g/100g).

Reviewer 2 Report

Comments and Suggestions for Authors

The manuscript entitled“Nutritional Variation on Periodically Harvested Shoots and Fruits of Lagenaria siceraria Landraces” In this study, We can learn that determining the nutritional status of shoots and fruits at different growth stages is essential for selecting local varieties for optimal harvesting and for meeting the required daily nutrient intake.

However, there are some fundamental errors, redundant parts, and incomplete statements in the manuscript. As follows:

1.     Some of the English expressions in the text are not appropriate, please make further corrections.

2.     Firstly, there is a lack of clarity about a harvesting criterion for the samples.

3.     Recommendations for improving the quality of illustrations.

4.     Did you measure this data and finally summarise the best harvesting periods for different varieties in different regions?

5.     The conclusion section should accurately express the results of the experiment, and it is recommended that the conclusion section be rewritten.

Comments on the Quality of English Language

  Some of the English expressions in the text are not appropriate, please make further corrections.

Author Response

Reviewer 2

General reviewer’s comments

Item

Comment

Response

Page 43, line 1150 - 1156

Firstly, there is a lack of clarity about a harvesting criterion for the samples.

Corrected as - For shoot nutrient composition , ten shoot tips (three-leaved shoots) per plot were harvested for each landrace at 42, 49, 56, and 63 days after sowing (DAS). For fruit nutrient composition, ten fruits per plot were harvested at 7, 14, 21 days after anthesis (DAA) for each landrace. Samples (shoots and fruits) harvested from each plot constituted a replicate (n = 3), meaning for shoots; 10×3 = 30 shoots per landrace and for fruits; 10×3 = 30 fruits per landrace from triplicate plots were used for analysis.

Page 34 – 35, line 766 – 798

Recommendations for improving the quality of illustrations.

The illustrations (Figure 2 and 3) have been enlarged for improved quality.

Did you measure this data and finally summarize the best harvesting periods for different varieties in different regions?

The landraces from different origins were grown on the same field. We then prepared the samples for analysis, recorded and interpreted the data thus summarizing the best harvesting period for shoots and fruits among different landraces based on their nutrient composition.

Page 46, line 1255 – 1283

The conclusion section should accurately express the results of the experiment, and it is recommended that the conclusion section be rewritten.

Corrected as - The investigated Lagenaria siceraria landraces had notable variations in their nutritional composition in sequentially harvested shoots (42–63 DAS) and fruits (7–21 DAA). Landraces KRI and DSI at 42–63 DAS, produced shoots of higher nutritional value while fruits of ESC and KSC at 7–21 DAA were the most nutritious among all landraces. Most nutrient traits much like calcium exhibited a progressive increase with shoot and fruit maturity, emphasizing the significance of mineral element mobility on the availability of nutrients on various plant parts. Nutrient traits of similar chemical properties and function in plants correlated positively with each other such as Ca and Mg, Mg with K, as well as K and N. The micronutrients (Ca, Mg, K, P, and N) in shoots were the main contributors to variability whereas in fruits macronutrients (Fe, Zn, Cu, and Al) contributed more to the variation based on the principal component analysis. The proximate composition also differed with maturity, where the majority of landraces had lower ADF, NDF, and ADL, though they had high carbohydrates at the juvenile (42–49 DAS) stages compared to mature stages at 56–63 DAS. Hence, L. siceraria plant parts consumed early will be highly digestible and have a high caloric value. The biplot and dendrogram grouped landraces with similar nutrient attributes that also correlate positively with each other. The study recorded for the first time the variation on nutrient composition in sequentially harvested shoots and fruits of L. siceraria landraces across different growth stages. Thus, providing valuable insights into the nutritional dynamics of L. siceraria landraces across various growth stages, offering a basis for understanding and optimizing for their most nutritive growth phase.

Some of the English expressions in the text are not appropriate, please make further corrections.

The manuscript was proofread thoroughly, and language errors were corrected where possible.

Round 2

Reviewer 2 Report

Comments and Suggestions for Authors

The manuscript entitled “Nutritional Variation on Periodically Harvested Shoots and Fruits of Lagenaria siceraria Landraces” focuses on 19 varieties of L. siceraria and uses ICP-OES method to determine various nutritional components in the buds and fruits of plants. The results showed significant differences in nutritional content among different varieties of L. siceraria at different growth stages. This result provides a reference for understanding the nutritional dynamics of different L. siceraria varieties at different growth stages. This study involves a variety of L. siceraria varieties and has certain practical application value. However, most of the content in this article is too redundant and lacks emphasis on key points. At the same time, most of the data only appears in tables and has not been created into beautiful images. In addition, there are still some issues that need to be modified. The specific opinions are as follows:

Line 77: Photos of 19 varieties of L. siceraria fruits and seeds should be added here.

Line 85: The content of section “2.1. Macronutrient Composition” is lengthy. It is recommended to subdivide this section and break it down into several headings.

Line 139: The full name of the abbreviation is marked when it first appears, and abbreviations are used uniformly in subsequent content. “DAA” is not appearing here for the first time. Check the entire text and correct similar errors.

Line 259: The content of section “2.2. Micronutrient Composition” is lengthy. It is recommended to subdivide this section and break it down into several headings.

Line 656: The correlation analysis results are presented more intuitively through graphs, and relevant data can be placed in attachments.

Line 724: The results of principal component analysis are presented more intuitively through graphs, and the data is placed in the attached file.

Line 839: The content of the “Discussion” section needs to be concise in language and highlight key points.

Line 1293: The “Conclusions” section is usually followed by the “Discussion” section. The “Materials and Methods” section is either placed after the introduction or at the end of the paper.

Author Response

Reviewer 2

General reviewer’s comments

Item

Comment

Response

1

Line 77: Photos of 19 varieties of L. siceraria fruits and seeds should be added here.

Photos  of fruits and seeds discussed in Table 1 have been included.

2

Line 85: The content of section “2.1. Macronutrient Composition” is lengthy. It is recommended to subdivide this section and break it down into several headings.

The manuscript has been separated into two manuscripts to eliminate its lengthy and winding nature. “Nutrient Composition of Lagenaria siceraria Fruits at Different Growth Stages” and “Nutritional Composition of Lagenaria siceraria Shoots Harvested at Different Growth Stages” are the titles of the now separated manuscript.

3

Line 139: The full name of the abbreviation is marked when it first appears, and abbreviations are used uniformly in subsequent content. “DAA” is not appearing here for the first time. Check the entire text and correct similar errors.

Full names of  all abbreviations have been removed after the first appearance throughout both manuscripts document.

4

Line 259: The content of section “2.2. Micronutrient Composition” is lengthy. It is recommended to subdivide this section and break it down into several headings.

The manuscript has been separated into two manuscripts to eliminate its lengthy and winding nature. “Nutrient Composition of Lagenaria siceraria Fruits at Different Growth Stages” and “Nutritional Composition of Lagenaria siceraria Shoots Harvested at Different Growth Stages” are the titles of the now separated manuscript.

5

Line 656: The correlation analysis results are presented more intuitively through graphs, and relevant data can be placed in attachments.

Relevant data has been provided in the attachments.

6

Line 724: The results of principal component analysis are presented more intuitively through graphs, and the data is placed in the attached file.

 Relevant data has been provided in the attachments.

7

Line 839: The content of the “Discussion” section needs to be concise in language and highlight key points.

 The discussion section has been revised for language conciseness to the best of our abilities while highlighting the key findings and their relatedness to comparable studies of similar plant species and/or fruit classes.

8

Line 1293The “Conclusions” section is usually followed by the “Discussion” section. The “Materials and Methods” section is either placed after the introduction or at the end of the paper.

An article submission template for the year 2024 provided by Plants was used to structure and align the manuscript in this order (Introduction, Results, Discussion, Materials & Methods, and Conclusion)

Round 3

Reviewer 2 Report

Comments and Suggestions for Authors

The manuscript entitled“Nutritional Variation on Periodically Harvested Shoots and Fruits of Lagenaria siceraria Landraces” In this study, We can learn that determining the nutritional status of shoots and fruits at different growth stages is essential for selecting local varieties for optimal harvesting and for meeting the required daily nutrient intake.

However, there are some fundamental errors, redundant parts, and incomplete statements in the manuscript. As follows:

1. Some of the English expressions in the text are not appropriate, please make further corrections.

2. Firstly, there is a lack of clarity about a harvesting criterion for the samples.

3. Recommendations for improving the quality of illustrations.

Comments on the Quality of English Language

need further English edits

Author Response

General reviewer’s comments

Item

Comment

Response

1

Some of the English expressions in the text are not appropriate, please make further corrections

English expressions in the text have been corrected to the best of our abilities and they are available in the documents with tract changes.

2

Firstly, there is a lack of clarity about a harvesting criterion for the samples.

The harvesting criterion of fruits and shoots has been clarified as follows:

·       Fruits were harvested at intervals of 7, 14, and 21days after anthesis (DAA). In each landrace, 10 fruits were harvested from different plants within each plot. Therefore, fruits from one plot constituted a replicate (n = 3).

·       Shoot tips with a minimum of three fully opened leaves were harvested at intervals of 42, 49, 56, and 63 days after sowing (DAS). Ten shoot tips were harvested from different plants within each plot. Therefore, shoots harvested from each plot constituted a replicate (n = 3).

3

Recommendations for improving the quality of illustrations

Figures 1 and 2 have been expanded on a landscape page layout to provide a more detailed picture. A greater quality of image(s) is not available because the harvested fruits were processed for nutrient composition on the same day of harvest.